# Hyperbolic RQ-VAE enhanced Generative Recommendation with Differential-Length Codebook Strategy

Aoran Zhang[1]   Yu-Bin Yang[1]   Yonghong Yu[1][2]

## Abstract

Recently, the integration of large language models (LLMs) with generative recommendation (GR) has demonstrated promising potential. However, most existing GR methods adopt residual quantization to implicitly model hierarchical relationships across codebook layers in Euclidean space, which distorts the intrinsic tree-like hierarchy and leads to low codebook utilization. To address these issues, we propose a Hyperbolic RQ-VAE enhanced Generative Recommendation, namely HG-Rec. Specifically, HG-Rec enhances the residual quantization mechanism by embedding the latent discrete representations into hyperbolic space to explicitly model hierarchical relationships across codebook layers. Motivated by the exponential volume growth of hyperbolic space, we further design a differential-length codebook strategy, i.e. the codebook size follows a pyramidal structure, which aligns with the tree-like structure and effectively compresses the codebook size. Hence, benefiting from the alignment of hyperbolic geometry and codebook hierarchy, HG-Rec achieves lower collision rates, more uniform codebook usage, and less training time compared to existing methods. Extensive experiments across multiple benchmark datasets demonstrate that HG-Rec consistently achieves state-of-the-art performance. The code is available in https://github.com/zar123123/HG-Rec.

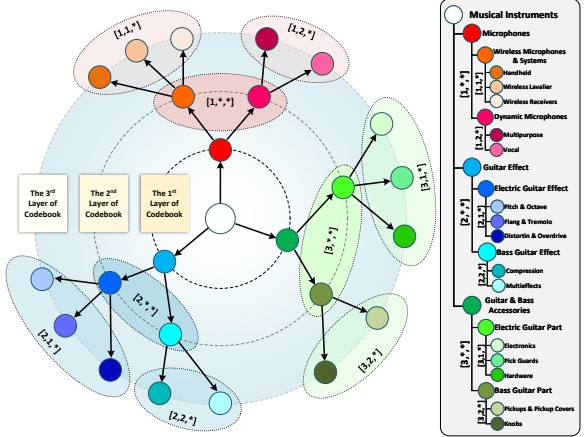

*Figure 1.* The case study of RQ-VAE on Instruments. Each item is quantized as a discrete codeword tuple $(c_1, c_2, c_3)$, where $c_\ell$ is the codeword selected from the $\ell^{th}$ layer. And $(c_1, *, *)$ indicates that only the first-layer codeword is fixed. Moreover, RQ-VAE introduces a coarse-to-fine hierarchical structure, where the first layer codeword $c_1$ corresponds to coarse-level category, while second/third layer codeword $c_2/c_3$ correspond to fine-grained levels.

## 1. Introduction

Nowadays, recommendation systems (Wu et al., 2024; Zhang et al., 2025c) have become a key component of various application platforms, enabling personalized and adaptive content delivery tailored to users' preferences. Traditional recommendation systems (Rendle et al., 2009; He et al., 2020; Sun et al., 2021) focus on discrete item IDs and shallow user–item interactions, which struggle to capture high-level semantics.

Owing to the powerful capability of LLMs (OpenAI, 2024), many researchers (Geng et al., 2022; Liu et al., 2023; Zhang et al., 2025d) have incorporated them into recommendation tasks, thereby fostering the emergence of a GR paradigm. Specifically, GRs (Rajput et al., 2023; Zheng et al., 2024; Zhai et al., 2024) typically assign each item with a unique identifier and employ LLMs to directly generate the identifiers of next items based on users' interaction history. The process of representing items as LLM-readable identifiers (i.e. item tokenization) can be viewed as the bridge be-

[1]State Key Laboratory of Novel Software Technology, Nanjing University, Nanjing 210023, China [2]College of Tongda, Nanjing University of Posts and Telecommunications, Yangzhou 225127, China. Correspondence to: Yu-Bin Yang <yangyubin@nju.edu.cn>.

*Proceedings of the 43rd International Conference on Machine Learning*, Seoul, South Korea. PMLR 306, 2026. Copyright 2026 by the author(s).

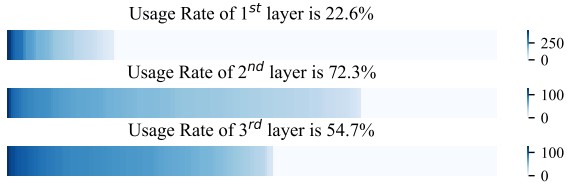

*Figure 2.* The codewords usage of RQ-VAE with codebook size [256,256,256] on Instruments.

tween the semantic space of the LLMs and the discrete item space, enabling the recommendation models to generate target items in a sequence-to-sequence manner without relying on traditional ranking structures.

The existing item tokenization techniques can be broadly divided into three main categories, i.e. ID-based (Hua et al., 2023), context aware-based (Hou et al., 2025; Zhong et al., 2025) and codebook-based tokenizations (Zheng et al., 2024; Wang et al., 2024). Unlike ID-based and context aware-based tokenizations, codebook-based tokenization utilizes RQ-VAE (Singh et al., 2024) to encode items into hierarchical token sequences, which naturally aligns with the sequence generation mechanism of LLMs. As shown in Figure 1, the inter-layer relationships of codebook exhibit a clear hierarchical structure. Essentially, RQ-VAE introduces a coarse-to-fine hierarchical structure in the latent space via residual quantization, which is graph-isomorphic to the tree structure. However, existing codebook-based tokenization strategies implicitly model the hierarchical relationships across codebook layers in Euclidean space, which impose structural limitations when representing the inherently hierarchical relationships among codebook layers. Furthermore, the above geometric mismatch distorts hierarchical relationships, leading to imbalanced codewords utilization. As a result, only a part of codewords is frequently used, while a portion of the codebook remains rarely or never activated, which is aligned with the statistics in Figure 2.

Recently, hyperbolic space (Peng et al., 2022; Desai et al., 2023; Zhang et al., 2025a) has attracted increasing attention for modeling hierarchical/tree-like structures, due to its properties of exponential volume growth and low-distortion embedding. This naturally motivates us to explore hyperbolic geometry for modeling the hierarchical codebook structure induced by codebook-based tokenization, and propose a hyperbolic RQ-VAE enhanced generative recommendation, namely HG-Rec. Specifically, HG-Rec includes two important designs: **(1) Hyperbolic RQ-VAE** enhances the residual quantization mechanism of RQ-VAE via embedding the latent discrete representation into hyperbolic space, which effectively captures the hierarchical relationships across codebook layers. To ensure stable optimization in hyperbolic space, we compute all residual updates in the tangent space. **(2) Differential-length codebook strategy** intro-

duces the growth rate of hyperbolic space to align with the tree-like structure and effectively compress the codebook size. Furthermore, we systematically evaluate the advantages of HG-Rec compared to existing methods in terms of collision rate, codebook usage and training time. The contributions are summarized as follows:

- We propose HG-Rec, a hyperbolic RQ-VAE enhanced generative recommendation model, which effectively captures the hierarchical relationships across codebook layers.

- HG-Rec adopts hyperbolic RQ-VAE to construct a hierarchical codebook and ensure stable optimization. Moreover, HG-Rec utilizes a differential-length codebook strategy, i.e. the codebook size follows a pyramidal structure, to effectively align with the tree-like hierarchy and compress the codebook size.

- We empirically validate three advantages (lower collision rate, more uniform codebook usage, and less training time) of HG-Rec, and conduct extensive experiments on three public datasets to demonstrate the effectiveness of HG-Rec.

## 2. Preliminaries

### 2.1. Problem description

The sequential recommendation system contains two fundamental entities, i.e. the set of users $U$ and the set of items $I$. The items $i \in I$ that are visited by a user $u \in U$ in chronological order are represented as a sequence $S_u = \{i_1, i_2, \cdots, i_t\}$, where $t$ is the number of actions. And the input item sequences of all users can be written as $A = \{S_1, S_2, \cdots, S_{|U|}\}$. In this paper, we design a codebook-based tokenizer to map the input item sequences $A$ into token sequences $C = \{C_1, C_2, \cdots, C_{|U|}\}$. Based on the resulting token sequences, we train a GR model to generate the next tokens, which can be reconstructed as the next candidate item $\hat{i}_{t+1}$.

### 2.2. Hyperbolic geometry

In hyperbolic geometry, the Poincaré ball model (Nickel & Kiela, 2017) is a classical and widely used instantiation of hyperbolic space. The Poincaré model $\mathbb{B}$ is a manifold with a Riemannian metric $g_c^{\mathbb{B}}$. $\mathbb{B}_c^n = \left\{ x \in \mathbb{R}^n : \|x\| < \frac{1}{\sqrt{c}} \right\}$ is the open $n$-dimensional unit ball with the curvature $c$, where $\|\cdot\|$ denotes the Euclidean norm. Riemannian metric $g_c^{\mathbb{B}}(x) = \left( \frac{2}{1-c\|x\|^2} \right)^2 g^{\mathbb{E}}$, and $g^{\mathbb{E}} = I_n$ denotes the Euclidean metric tensor. For any two points $\mathbf{x}, \mathbf{y} \in \mathbb{B}_c^n$, the

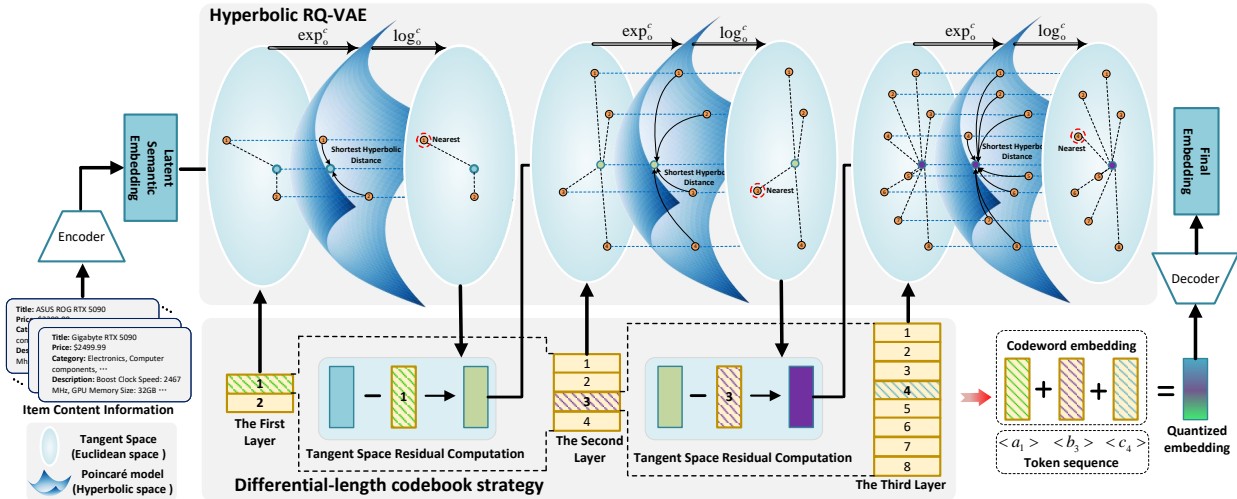

*Figure 3.* The overall framework of our proposed HG-Rec. HG-Rec includes two important components, i.e. Hyperbolic RQ-VAE and Differential-length codebook strategy.

hyperbolic distance between them can be defined as follows,

$$d_{\mathbb{B}}(\mathbf{x}, \mathbf{y}) = \frac{2}{\sqrt{c}} \operatorname{artanh} \left( \sqrt{c} \| (-\mathbf{x}) \oplus_c \mathbf{y} \| \right), \quad (1)$$

where $\oplus_c$ is Möbius addition, formally,

$$\mathbf{x} \oplus_c \mathbf{y} = \frac{(1 + 2c\langle\mathbf{x}, \mathbf{y}\rangle + c\|\mathbf{y}\|^2)\mathbf{x} + (1 - c\|\mathbf{x}\|^2)\mathbf{y}}{1 + 2c\langle\mathbf{x}, \mathbf{y}\rangle + c^2\|\mathbf{x}\|^2\|\mathbf{y}\|^2},$$
$$\langle\mathbf{x}, \mathbf{y}\rangle := \mathbf{x}^\top \mathbf{y}. \quad (2)$$

When $c \to 0$, the Möbius add (Chami et al., 2019; Liu et al., 2019) degrades into Euclidean addition. $d_{\mathbb{B}}(\mathbf{x}, \mathbf{y})$ grows exponentially near the boundary, naturally fitting the geometric characteristics of tree and hierarchical structures. Moreover, for each point $\mathbf{x} \in \mathbb{B}_c^n$, there is a tangent space $\mathcal{T}_x^c \mathbb{B}_c^n$. Considering that the map function is simple and symmetric at the origin point $\mathbf{o}$, we set the target point $x$ as the origin. For $\mathbf{v} \in \mathcal{T}_\mathbf{o}^c \mathbb{B}_c^n$ and $\mathbf{h} \in \mathbb{B}_c^n$, the Poincaré model defines the exponential map function $\exp_\mathbf{o}^c : \mathcal{T}_\mathbf{o}^c \mathbb{B}_c^n \to \mathbb{B}_c^n$, which is used to map point $\mathbf{v}$ into hyperbolic space. Formally,

$$\exp_\mathbf{o}^c(\mathbf{v}) = \frac{1}{\sqrt{c}} \tanh(\sqrt{c}\|\mathbf{v}\|) \frac{\mathbf{v}}{\|\mathbf{v}\|}, \quad (3)$$

where $\tanh(\mathbf{x}) = \frac{e^\mathbf{x} - e^{-\mathbf{x}}}{e^\mathbf{x} + e^{-\mathbf{x}}}$ is the hyperbolic tangent function. In order to map points from hyperbolic space to the corresponding target tangent space, the Poincaré model defines the logarithmic map function $\log_\mathbf{o}^c : \mathbb{B}_c^n \to \mathcal{T}_\mathbf{o}^c \mathbb{B}_c^n$,

$$\log_\mathbf{o}^c(\mathbf{h}) = \frac{1}{\sqrt{c}} \operatorname{artanh}(\sqrt{c}\|\mathbf{h}\|) \frac{\mathbf{h}}{\|\mathbf{h}\|}, \quad (4)$$

where $\operatorname{arctanh}(\mathbf{x}) = \frac{1}{2} \ln \left( \frac{1+\mathbf{x}}{1-\mathbf{x}} \right)$ is the inverse hyperbolic tangent function. In addition, for an arbitrary point $\mathbf{x} \in$ $\mathbb{B}_c^n$, $\mathbf{x} \neq \mathbf{o}$, the maps can be obtained via parallel transport or by using Möbius subtraction.

## 3. Method

In this section, we provide a detailed description of our proposed HG-Rec and the overall framework is illustrated in Figure 3. Motivated by

**Theorem 3.1.** *Assuming that RQ-VAE employs $L$ layers of residual quantization, with each layer having a codebook of size $K$. The residual quantization process induces a hierarchical structure on the latent space $\mathbb{R}^n$, which is graph-isomorphic to a rooted $K$-ary tree of depth $L$.(Proof in Appendix B.1)*

We focus on modeling the hierarchical relationships across codebook layers via hyperbolic RQ-VAE in Section 3.1. Moreover, to effectively align with the tree-like hierarchy and compress the codebook size, HG-Rec introduces the growth rate of hyperbolic space to differential-length codebook strategy in Section 3.2. Finally, we describe the model training and inference process in Section 3.3.

### 3.1. Hyperbolic RQ-VAE

To explicitly capture the hierarchical relationships across codebook layers, we propose hyperbolic RQ-VAE, which encodes item semantics into discrete token sequences via residual quantization in hyperbolic space. Especially, we enhance the residual quantization mechanism of RQ-VAE via embedding the latent discrete representation into a hyperbolic manifold. To ensure stable and efficient optimization

in hyperbolic space, we compute all residual updates in the tangent space under a constant-curvature geometry. This design endows the codebooks with exponentially capacity and a naturally hierarchical organization, allowing HG-Rec to effectively encode fine-grained semantic distinctions across quantization layers.

**Latent semantic embedding extraction.** Given an item and its content information (i.e. titles, prices, categories and description), we have access to a pre-trained content extractor, e.g. Sentence-T5 (Ni et al., 2022) or LLaMA-7B (Touvron et al., 2023), to generate the corresponding semantic embedding $s$. Then, the semantic embedding $s$ is compressed into the latent semantic embedding $\mathbf{z} = \text{Encoder}(s)$, $\mathbf{z} \in \mathbb{R}^n$ via an encoder. The latent semantic embedding $\mathbf{z}$ is further quantized into a token sequence using a hyperbolic residual vector quantization process through $L$-layer codebooks. For each layer $\ell \in \{1, 2, \ldots, L\}$, we have a codebook $\mathcal{C}_\ell = \{\mathbf{e}_{\ell,1}, \ldots, \mathbf{e}_{\ell,K}\}$, where $K$ is the codebook size and $\mathbf{e}_{\ell,*}$ is the learnable codeword embedding.

**Hyperbolic residual quantization.** At the zero-th layer, the initial residual is $\mathbf{r}_0 = \mathbf{z}$, where $\mathbf{r}_0$ and $\mathbf{z}$ are defined in tangent space $\mathcal{T}_\mathbf{o}^c \mathbb{B}_c^n$, which is isomorphic to the Euclidean space $\mathbb{R}^n$. To explicitly capture the hierarchical relationships across codebook layers, it is necessary to evaluate hyperbolic distances between the initial residual $\mathbf{r}_0$ (i.e. the latent semantic embedding $\mathbf{z}$) and the codebook entities $\mathbf{e}_{1,*}$ in first layer of codebooks. Since hyperbolic distances are only well-defined on the hyperbolic manifold, we first map both $\mathbf{r}_0 := \mathbf{z}$ and $\mathbf{e}_{1,*}$ from the tangent space into the hyperbolic space via the exponential map at the origin $\mathbf{o}$. Formally,

$$\mathbf{r}_{\ell-1}^{\mathbb{H}} = \exp_\mathbf{o}^c(\mathbf{r}_{\ell-1}), \ \mathbf{e}_{\ell,*}^{\mathbb{H}} = \exp_\mathbf{o}^c(\mathbf{e}_{\ell,*}), \qquad (5)$$

where $\exp_\mathbf{o}^c(\cdot)$ denotes the exponential map associated with the Poincaré model of curvature $-c$ and $\ell$ is the layer of codebook. Then, the selected codeword index $c_\ell$ with the shortest hyperbolic distance is defined as

$$c_\ell = \arg\min_i d_{\mathbb{B}}(\mathbf{r}_{\ell-1}^{\mathbb{H}}, \mathbf{e}_{\ell,i}^{\mathbb{H}}). \qquad (6)$$

Since subtraction is complex and may cause unstable parameter updates, we compute the residuals in the tangent space as follows,

$$\begin{cases} \mathbf{r}_{\ell-1} = \log_\mathbf{o}^c(\mathbf{r}_{\ell-1}^{\mathbb{H}}), \ \mathbf{e}_{\ell,*} = \log_\mathbf{o}^c(\mathbf{e}_{\ell,*}^{\mathbb{H}}), \\ \mathbf{r}_\ell = \mathbf{r}_{\ell-1} - \mathbf{e}_{\ell,c_\ell}, \end{cases} \qquad (7)$$

where $\log_\mathbf{o}^c(\cdot)$ denotes the logarithmic map at the origin of the Poincaré ball. This design decouples similarity evaluation (performed in hyperbolic space) from residual computation (performed in the tangent space), enabling both hierarchical structure modeling and stable optimization.

**Theorem 3.2.** *The exponential map function* $\exp_\mathbf{o}^c$ : $\mathcal{T}_\mathbf{o}^c \mathbb{B}_c^n \to \mathbb{B}_c^n$ *is well-defined. (Proof in Appendix B.2)*

**Theorem 3.3.** *The logarithmic map function* $\log_\mathbf{o}^c : \mathbb{B}_c^n \to \mathcal{T}_\mathbf{o}^c \mathbb{B}_c^n$ *is well-defined. (Proof in Appendix B.3)*

Due to **Theorem 3.2** and **Theorem 3.3**, the well-definedness of exponential and logarithmic map functions guarantees that all Euclidean vectors can be uniquely and smoothly mapped into the Poincaré ball and mapped back to the tangent space. This ensures that hyperbolic distance computation and residual updates are geometrically valid and numerically stable.

**The loss of hyperbolic RQ-VAE.** When we have $L$-layer codebook, the quantized embedding of $\mathbf{z}$ can be obtained according to $\hat{\mathbf{z}} = \sum_{\ell=0}^L \mathbf{e}_{\ell,c_\ell}$, where the computation of quantized embedding is in the tangent space. The quantized embedding $\hat{\mathbf{z}}$ will decode to the reconstructed semantic embedding $\hat{s}$. The overall loss function of hyperbolic RQ-VAE is formalized as

$$\begin{cases} s^{\mathbb{H}} = \exp_\mathbf{o}^c(s), \ \hat{s}^{\mathbb{H}} = \exp_\mathbf{o}^c(\hat{s}), \\ \mathbb{L}_{recon}^{\mathbb{H}} = d_{\mathbb{B}}(s^{\mathbb{H}}, \hat{s}^{\mathbb{H}})^2, \\ \mathbb{L}_{rqvae}^{\mathbb{H}} = \sum_{\ell=1}^L d_{\mathbb{B}}(\text{sg}[\mathbf{r}_{\ell-1}^{\mathbb{H}}], \mathbf{e}_{\ell,c_\ell}^{\mathbb{H}})^2 + \beta \cdot d_{\mathbb{B}}(\mathbf{r}_{\ell-1}^{\mathbb{H}}, \text{sg}[\mathbf{e}_{\ell,c_\ell}^{\mathbb{H}}])^2, \\ \mathbb{L}^{\mathbb{H}} = \mathbb{L}_{recon}^{\mathbb{H}} + \mathbb{L}_{rqvae}^{\mathbb{H}}, \end{cases}$$
$$(8)$$

where $\text{sg}[\cdot]$ represents the stop-grad operation, and $\beta$ is a loss coefficient. In $\mathbb{L}^{\mathbb{H}}$, both reconstruction loss $\mathbb{L}_{recon}^{\mathbb{H}}$ and the residual quantization loss $\mathbb{L}_{rqvae}^{\mathbb{H}}$ are defined using hyperbolic distance to ensure geometric consistency with the underlying representation space. Although the embeddings are parameterized in the tangent space for numerical stability, all semantic similarity and hierarchical relations are defined in hyperbolic space. Therefore, utilizing Euclidean reconstruction loss and residual quantization loss may introduce a geometric mismatch. For geometric consistency, we uniformly compute all loss terms in hyperbolic space.

### 3.2. Differential-length codebook strategy

Traditional codebook sizes are typically set to $[256, 256, 256]$, which implies that the total number of possible combinations formed by selecting one codeword from each layer is $256^3$. For recommendation systems, this configuration provides a theoretical representational capacity of approximately $10^8$ items, which is often significantly larger than the actual number of items in real-world datasets. In practice, widely used recommendation datasets typically contain about $10^5$ items, indicating that even with zero collision rate, the utilization rate of the codebook is only 1‰. Moreover, hyperbolic RQ-VAE essentially quantifies the item semantic embeddings into coarse-to-fine representations, i.e. earlier layers capture coarse-grained semantics and later layers encode increasingly fine-grained

variations. Consequently, fine-grained representations exhibit higher semantic diversity and demand substantially larger representational capacity than coarse-grained representations.

**Theorem 3.4.** *The geometric capacity of hyperbolic space grows exponentially.*

*Proof.* According to Riemannian geometry, the volume element in a Poincaré ball is

$$dV_{\mathbb{B}}(\mathbf{x}) = \left( \frac{2}{1 - c\|\mathbf{x}\|^2} \right)^n d\mathbf{x}. \qquad (9)$$

Integrating the volume element in polar coordinates as follows,

$$\text{Vol}_{\mathbb{B}}(\rho) = \omega_{n-1} \int_0^\rho \left( \sinh(\sqrt{c}t) \right)^{n-1} dt, \qquad (10)$$

where $\omega_{n-1}$ denotes the surface area constant of the unit $(n-1)$-dimensional Euclidean sphere, $t$ is geodesic distance and $\rho$ is geodesic radius. When $t \to \infty$, $\sinh(\sqrt{c}t) \sim \frac{1}{2}e^{\sqrt{c}t}$ and $(\sinh(\sqrt{c}t))^{n-1} \sim \left(\frac{1}{2}\right)^{n-1} e^{(n-1)\sqrt{c}t}$. Hence, $\text{Vol}_{\mathbb{B}}(r) \sim e^{(n-1)\sqrt{c}\rho}$. $\qquad\square$

According to **Theorem 3.4**, in hyperbolic space, regions closer to the origin have smaller representational capacity, while regions closer to the boundary exhibit significantly larger capacity. This property naturally aligns with the coarse-to-fine hierarchical structure induced by residual quantization. Hence, assigning identical codebook sizes to all quantization layers leads to a structural mismatch, i.e. coarse-grained layers are over-parameterized, while fine-grained layers suffer from insufficient representational capacity. To address above issue, we utilize the volume growth rate of hyperbolic space to define our proposed differential-length codebook as follows,

$$K_\ell \propto K_1 \cdot e^{(n-1)\sqrt{c}\rho}, \qquad (11)$$

where $K_1$ is the size of first layer codebook and $\rho$ is geodesic radius. From Eq. (11), we can observe that the growth rate of capacity is controlled by the radius $r$ and $e^{(n-1)\sqrt{c}}$ is a fixed value. Set the step size $\Delta\rho$ between adjacent layers, we have $r_\ell = r_0 + \ell\Delta\rho$. And the growth rate $\gamma \triangleq e^{(n-1)\sqrt{c}\Delta\rho}$, we can draw $K_\ell \propto e^{(n-1)\sqrt{c}r_\ell} \propto e^{(n-1)\sqrt{c}\ell\Delta\rho}$. Hence, Eq. (11) can be rewritten as

$$K_\ell \propto K_1 \cdot \gamma^\ell. \qquad (12)$$

In practice, in order to effectively compress the codebook, we guarantee that the size of the outermost layer is not larger than 256. Hence, we choose $\gamma \approx 2$ and $K_1 \in \{16, 32, 64\}$. And our proposed differential-length codebook strategy, i.e. the codebook size follows a pyramidal structure, not only effectively compresses the codebook size, but also aligns with the hierarchical structure.

## 3.3. Model training and inference

We first train hyperbolic RQ-VAE to map item sequences $A$ into item token sequences $C$. The resulting item token sequences are then used in both the training and inference stages of the GR model.

For the training stage of GR model, taking $C^{in}$ as input, we train a Transformer encoder-decoder module (Raffel et al., 2020) to autoregressively generate $C^{out}$. The model parameters are optimized by adopting the negative log-likelihood loss over the target token sequence. Specifically, the negative log-likelihood loss is defined as

$$\mathbb{L} = -\sum_{u=1}^{|U|} \sum_{t=1}^{|C_u^{out}|} \log p(C_{u,t}^{out} \mid C_{u,<t}^{out}, C_u^{in}), \qquad (13)$$

where $C_{u,t}^{out}$ indicates that the $t^{th}$ token in $C_u^{out}$ of user $u$ and $C_{u,<t}^{out}$ represents the tokens before $C_{u,t}^{out}$. Moreover, $C_u^{in}$ is the input token sequence of user $u$.

For the inference stage of GR model, our goal is to generate the top-$N$ items that align with the preferences of a given user. To this end, the decoder employs beam search (Rajput et al., 2023) over the discrete index tokens.

## 3.4. Discussion

The process of residual quantization is essentially inducing a coarse-to-fine tree-like structure in latent space. However, traditional RQ-VAE embeds this structure in Euclidean space, which not only distorts the inherent hierarchy but also leads to inefficient representation. This is because the volume grows of Euclidean space only polynomially with respect to the radius, leading to crowding effects and insufficient separation between codewords at deeper levels. In contrast, our proposed hyperbolic RQ-VAE primarily improves the latent space of codebook via aligning the hierarchical structure and hyperbolic geometry. This enables the model to learn more discriminative embeddings, thereby generating higher-quality codebooks and effectively reducing the collision rates. The differential-length codebook strategy operates at the aspect of capacity allocation, which allows the model to allocate more capacity to fine-grained layers while avoiding redundancy in coarse layers. Therefore, the two proposed components enhance the traditional RQ-VAE from two different perspectives. Especially, there is no conflict between components, as differential-length codebook strategy is based on the growth rate of hyperbolic space volume, which naturally fits hyperbolic space.

## 4. Experiments

In this section, we conduct extensive experiments on three real-world datasets and further explore three intrinsic properties of HG-Rec.

*Table 1.* Performance Comparison on five datasets. The best results are in **bold** and the runner-up is underlined. $R@N$ and $N@N$ are short for $Recall@N$ and $NDCG@N$, respectively. *Improv.* denotes the percentage improvement of our method compared to the strongest baseline method.

| Methods | Beauty | | | | Instruments | | | | Yelp | | | |
|---|---|---|---|---|---|---|---|---|---|---|---|---|
| | $R@5$ | $R@10$ | $N@5$ | $N@10$ | $R@5$ | $R@10$ | $N@5$ | $N@10$ | $R@5$ | $R@10$ | $N@5$ | $N@10$ |
| Caser | 0.0208 | 0.0335 | 0.0132 | 0.0173 | 0.0521 | 0.0677 | 0.0379 | 0.0430 | 0.0132 | 0.0228 | 0.0081 | 0.0111 |
| HGN | 0.0405 | 0.0629 | 0.0243 | 0.0311 | 0.0781 | 0.0960 | 0.0654 | 0.0712 | 0.0193 | 0.0328 | 0.0118 | 0.0161 |
| SASRec | 0.0438 | 0.0650 | 0.0269 | 0.0330 | 0.0821 | 0.1080 | 0.0688 | 0.0740 | 0.0209 | 0.0361 | 0.0134 | 0.0183 |
| Bert4Rec | 0.0407 | 0.0653 | 0.0238 | 0.0308 | 0.0814 | 0.1034 | 0.0675 | 0.0745 | 0.0202 | 0.0341 | 0.0127 | 0.0173 |
| P5-RID | 0.0216 | 0.0470 | 0.0181 | 0.0281 | 0.0701 | 0.0823 | 0.0615 | 0.0657 | 0.0220 | 0.0321 | 0.0155 | 0.0188 |
| P5-IID | 0.0388 | 0.0624 | 0.0273 | 0.0336 | 0.0739 | 0.0871 | 0.0647 | 0.0692 | 0.0226 | 0.0385 | 0.0142 | 0.0192 |
| P5-TID | 0.0185 | 0.0426 | 0.0134 | 0.0250 | 0.0448 | 0.0633 | 0.0340 | 0.0400 | 0.0056 | 0.0084 | 0.0039 | 0.0048 |
| P5-SemID | 0.0440 | 0.0661 | 0.0294 | 0.0365 | 0.0776 | 0.0880 | 0.0706 | 0.0744 | 0.0197 | 0.0316 | 0.0128 | 0.0166 |
| P5-CID | 0.0481 | 0.0692 | 0.0312 | 0.0362 | 0.0891 | 0.1065 | 0.0770 | 0.0830 | 0.0241 | 0.0405 | 0.0161 | 0.0212 |
| TIGER | 0.0502 | 0.0775 | 0.0330 | 0.0418 | 0.0979 | 0.1214 | 0.0811 | 0.0886 | 0.0234 | 0.0384 | 0.0154 | 0.0203 |
| LC-Rec | 0.0519 | 0.0788 | 0.0338 | 0.0420 | 0.0935 | 0.1176 | 0.0779 | 0.0868 | 0.0227 | 0.0355 | 0.0148 | 0.0189 |
| Letter | 0.0533 | 0.0797 | 0.0355 | 0.0426 | 0.0982 | 0.1219 | 0.0811 | 0.0880 | 0.0239 | 0.0398 | 0.0161 | 0.0210 |
| ActionPiece | 0.0539 | 0.0816 | 0.0357 | 0.0440 | 0.0999 | 0.1245 | 0.0812 | 0.0901 | 0.0242 | 0.0414 | 0.0161 | 0.0219 |
| HG-Rec | **0.0572** | **0.0872** | **0.0377** | **0.0473** | **0.1058** | **0.1315** | **0.0862** | **0.0945** | **0.0275** | **0.0445** | **0.0180** | **0.0233** |
| *Improv.* | 6.2% | 6.8% | 5.5% | 7.4% | 5.9% | 5.6% | 6.2% | 4.8% | 13.5% | 7.5% | 11.8% | 6.3% |

## 4.1. Experimental settings

**Datasets:** We choose three real-world datasets, i.e. Beauty, Instruments and Yelp, from different domains to evaluate the effectiveness of HG-Rec. Detailed datasets information are shown in Appendix C.

**Baselines:** We select the following methods as baselines: (1) Traditional sequential recommendation methods: Caser (Tang & Wang, 2018), HGN (Ma et al., 2019), SASRec (Kang & McAuley, 2018) and Bert4Rec (Sun et al., 2019). (2) GR with ID-based tokenization: P5-RID, P5-IID, P5-TID, P5-SemID and P5-CID (Hua et al., 2023). (3) GR with context aware-based tokenization: ActionPiece (Hou et al., 2025). (4) GR with codebook-based tokenization: TIGER (Rajput et al., 2023), LC-Rec (Zheng et al., 2024) and LETTER (Wang et al., 2024). The detailed description is provided in Appendix D.

**Evaluation metrics:** We use $Recall@N$ and $NDCG@N$ as metrics to evaluate the performance of all compared methods, where $N \in \{5, 10\}$.

**Implementation details:** The implementation details are shown in Appendix F.

## 4.2. Overall performance

The performance comparison between our proposed HG-Rec and the competitive baselines is presented in Table 1. In most cases, GR models are superior to traditional sequential recommendation methods, confirming the effectiveness of adopting the generative retrieval paradigm for recom-

mendation. Moreover, GR models with codebook-based tokenization generally outperform those adopting ID-based tokenization, as codebook-based approaches encode items into hierarchical token sequences that naturally align with the sequence generation mechanism of LLMs. In addition, ActionPiece achieves superior performance compared with all baseline methods, demonstrating that context-aware action tokenization is able to capture important sequence-level feature patterns that enhance recommendation performance.

On all datasets, our proposed HG-Rec consistently outperforms other methods. For example, in terms of $Recall@5$ and $NDCG@5$, HG-Rec improves ActionPiece by 6.2% and 5.5% on Beauty, 5.9% and 6.2% on Instruments, and 13.5% and 11.8% on Yelp, respectively. Different from GR methods, HG-Rec employs the hyperbolic RQ-VAE to explicitly capture the hierarchical relationships across codebook layers, and adopts a differential-length codebook strategy to compress the codebook size, effectively improving the performance of GR. Moreover, the parameter sensitivity analysis is presented in Appendix G.

## 4.3. Analysis of the codebook learning of HG-Rec

Since TIGER, LC-Rec and LETTER essentially adopt conventional RQ-VAE to generate codebook, we select vanilla RQ-VAE and our proposed hyperbolic RQ-VAE for further analysis. Specifically, we analyze the the advantages of hyperbolic RQ-VAE, i.e. collision rate (Section 4.3.1), codebook usage (Section 4.3.2), training time (Section 4.3.3) and hierarchical relationships (Section 4.3.4).

### 4.3.1. COMPARISONS OF COLLISION RATE

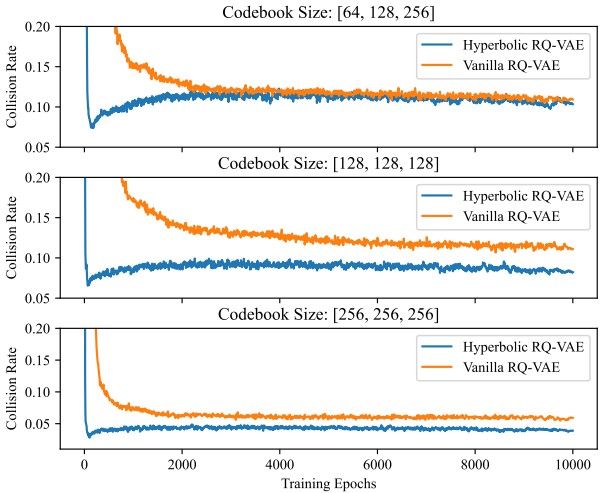

*Figure 4.* The training curve of Vanilla RQ-VAE and Hyperbolic RQ-VAE in terms of collision rates.

We compare the collision rates of vanilla RQ-VAE and hyperbolic RQ-VAE during training. Taking dataset Beauty as an example, we show the collision rates adopting differential-length codebook strategy (i.e. codebook size is [64, 128, 256]) and traditional codebook strategy (i.e. codebook size is [128, 128, 128] or [256, 256, 256]) in Figure 4. Moreover, the detailed experimental results are shown in Appendix H. Actually, we employ collision rate to measure whether the codebook is fully utilized. In general, a lower collision rate represents less redundancy and a more efficient use of the codebook. Compared to vanilla RQ-VAE, hyperbolic RQ-VAE has a lower collision rate. The results indicate that our proposed hyperbolic RQ-VAE is able to utilize the codebook more effectively and introduce a more diverse assignment of codewords. In other words, the geometric alignment between the hyperbolic space and the codebook space naturally enhances the model's capability of discrimination and capturing hierarchical relationship.

### 4.3.2. CODEBOOKS USAGE

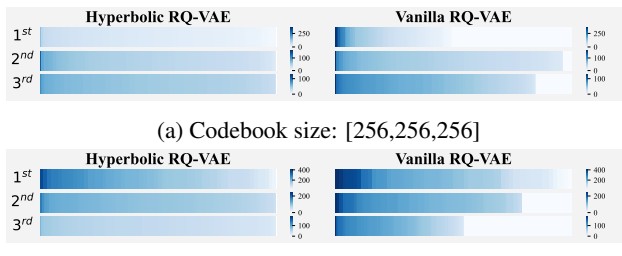

(a) Codebook size: [256,256,256]

(b) Codebook size: [64,128,256]

*Figure 5.* Codebooks usage on Beauty. Darker colors indicate higher usage frequency, while white denotes unused codewords.

In this section, we investigate the usage frequency distributions of codebooks that generated by vanilla RQ-VAE and hyperbolic RQ-VAE on Beauty. In Figure 5, the hyperbolic RQ-VAE achieves 100% codebook usage in all cases, while vanilla RQ-VAE exhibits lower and non-uniform usage. Interestingly, even if we reduce the codebook to a sufficiently small size, the vanilla RQ-VAE still fails to fully utilize all codewords. This phenomenon indicates that the vanilla RQ-VAE is prone to potential codeword collapse, which limits its representation capacity. In contrast, the hyperbolic RQ-VAE encourages a more balanced assignment of codewords, because it adopt hyperbolic distances to enhance the discrimination of latent features during quantization. The complete experiments in Appendix I.

### 4.3.3. TRAINING TIME

Similar to works (Rajput et al., 2023; Liu et al., 2025; Wei et al., 2025), we fully train both the vanilla RQ-VAE and the hyperbolic RQ-VAE, and record the number of epochs required to reach the minimum collision rate, along with the corresponding training time.

*Table 2.* The time comparisons of vanilla RQ-VAE and hyperbolic RQ-VAE under codebook size [256,256,256].

| Dataset | Methods | Per Epoch | Best Epoch | Total |
|---|---|---|---|---|
| Beauty | Vanilla RQ-VAE | 0.36s | $9740^{th}$ | 3724s |
| | Hyperbolic RQ-VAE | 0.60s | $165^{th}$ | 103s |
| Instruments | Vanilla RQ-VAE | 0.35s | $9200^{th}$ | 3907s |
| | Hyperbolic RQ-VAE | 0.50s | $380^{th}$ | 359s |
| Yelp | Vanilla RQ-VAE | 0.51s | $9880^{th}$ | 5445s |
| | Hyperbolic RQ-VAE | 0.82s | $110^{th}$ | 144s |

As reported in Table 2, hyperbolic RQ-VAE exhibits a higher per-epoch training time than vanilla RQ-VAE, because hyperbolic RQ-VAE introduces additional computations of hyperbolic distances and manifold mapping functions. Despite the high per-epoch cost, hyperbolic RQ-VAE only a small number of epochs to achieve the minimum collision rate, resulting in a short overall training time. For instance, compared to vanilla RQ-VAE, the overall training time of hyperbolic RQ-VAE is reduced by about 36×, 11×, and 38× on Beauty, Instruments, and Yelp, respectively. These results demonstrate the efficiency of hyperbolic RQ-VAE.

### 4.3.4. VISUALIZATION EXPERIMENT OF HIERARCHICAL RELATIONSHIPS

To further illustrate the hierarchical relationships across codebook layers, we visualize the embeddings of codewords learned by vanilla RQ-VAE and hyperbolic RQ-VAE. Specifically, all codewords embeddings are mapped into a hyperbolic space, where their positions are calculated by hyperbolic distances, as shown in Figure (6). The embed-

**Codebook size: [256, 256, 256]**

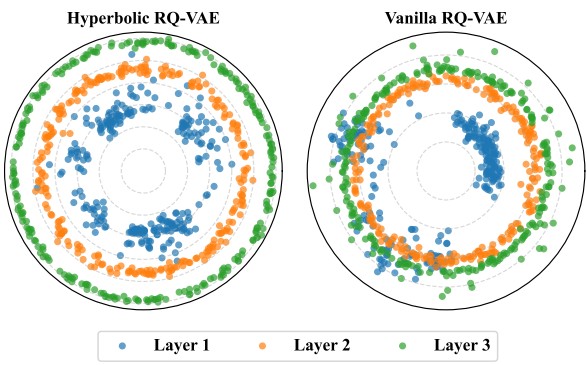

*Figure 6.* The visualizations of vanilla RQ-VAE and hyperbolic RQ-VAE on Beauty.

dings of both vanilla RQ-VAE and hyperbolic RQ-VAE exhibit hierarchical characteristics, which is consistent with **Theorem 3.1**. Compared to vanilla RQ-VAE, hyperbolic RQ-VAE demonstrates a more distinguishable hierarchy, characterized by a clear hierarchical distribution. This can be attributed to the natural geometric alignment makes hyperbolic RQ-VAE suitable for modeling hierarchical relationships. The detailed analysis is provided in Appendix J.

### 4.4. Ablation Study

We conduct an ablation study to further analyze the contribution of each component. The recommendation performance is measured using NDCG@10 and the detailed results are shown in Table 3.

*Table 3.* Ablation analysis of HG-Rec. The recommendation performance is measured using NDCG@10 and the best results are shown in **bold**.

| Variants | Beauty | Instruments | Yelp |
|---|---|---|---|
| *TIGER with traditional codebook strategy.* | | | |
| + [64,64,64] | 0.0408 | 0.0874 | 0.0197 |
| + [128,128,128] | 0.0410 | 0.0879 | 0.0201 |
| + [256,256,256] | 0.0418 | 0.0886 | 0.0203 |
| *HG-Rec w/o hyperbolic RQ-VAE (i.e. w/o H)* | | | |
| + [32,64,128] | 0.0412 | 0.0877 | 0.0203 |
| + [32,64,256] | 0.0427 | 0.0896 | 0.0208 |
| + [64,128,256] | 0.0424 | 0.0905 | 0.0209 |
| *HG-Rec w/o differential-length codebook strategy (i.e. w/o D)* | | | |
| + [64,64,64] | 0.0435 | 0.0902 | 0.0215 |
| + [128,128,128] | 0.0441 | 0.0908 | 0.0217 |
| + [256,256,256] | 0.0457 | 0.0931 | 0.0222 |
| HG-Rec | **0.0473** | **0.0945** | **0.0233** |

(1) To investigate the impact of codebook size on downstream recommendation task performance, we develop three variants of TIGER, each using traditional codebook strategy, where "+ [64, 64, 64]" denotes the model equipped

with a [64, 64, 64] codebook. We can observe that reducing the codebook size consistently leads to degraded model performance. The main reason is that the limited diversity of codewords selection leads to insufficient discriminative ability of recommendation models, which is consistent with the observations from Appendix G.1.

(2) To evaluate the effectiveness of differential-length codebook strategy, we remove the hyperbolic RQ-VAE module, resulting in a variant denoted as HG-Rec without hyperbolic RQ-VAE (i.e. *w/o H*). Essentially, *w/o H* can be regarded as TIGER utilizing differential-length codebook strategy. Compared to traditional codebook strategy, differential-length codebook strategy achieves better performance in most cases, which indicates that it is effective not only for HG-Rec but also for traditional codebook-based methods. This is because the differential-length codebook strategy cuts redundant and unused codewords at each layer, thereby simplifying the codeword assignment process and alleviating the difficulty of downstream recommendation.

(3) To verify the effectiveness of hyperbolic RQ-VAE, we drop the differential-length codebook strategy from HG-Rec (i.e. *w/o D*). Under the same codebook settings, the performances of *w/o D* consistently outperform those of TIGER. These improvements can be attributed to the natural alignment between the hyperbolic space and the codebook space. Moreover, *w/o D* employs hyperbolic RQ-VAE to generate the codebook, which effectively captures hierarchical relationships across codebook layers.

## 5. Related works

**Sequential Recommendation** aims to capture the sequential patterns among historical item sequences. Traditional sequential recommendation methods usually utilize Markov chain (Rendle et al., 2010; Feng et al., 2015) and Translation operation (He et al., 2017) to model sequential patterns. Recently, Sequential recommendation has evolved from traditional methods to deep learning-based approaches, i.e. GRU4Rec (Hidasi et al., 2016), Caser (Tang & Wang, 2018) and SelfGNN (Liu et al., 2024). Due to the effectiveness of contrastive learning techniques in alleviating the data sparsity problem, CL4SRec (Xie et al., 2021), ICLRec (Chen et al., 2022), DuoRec (Qiu et al., 2022) and CT4Rec (Zhang et al., 2025b) have extensively explored the integration of self-supervised learning with sequential recommendation by adopting different data augmentation strategies or contrastive learning frameworks. However, these methods overlook the rich item content information that can provide additional semantic context for representation learning.

**Generative recommendations** inherit the powerful capabilities of LLMs, including strong semantic understanding (Hariharan, 2025), reasoning (Huang & Chang, 2023) and

generalization (Yang et al., 2025). The existing GRs based on item tokenization can be broadly divided into three categories: ID-based, context aware-based, and codebook-based tokenization. SemID and CID (Hua et al., 2023), as early methods employing ID-based tokenizations, encode semantic information and collaborative signals respectively for item representations. RecSysLLM (Chu et al., 2023) introduces a novel mask mechanism for token sequences to inject entity knowledge into the LLM. Moreover, context aware-based tokenizations consider contextual relationships across all token sequences, and representative works include ActionPiece (Hou et al., 2025) and Pctx (Zhong et al., 2025). In addition, Codebook-based tokenizations adopt learnable codebooks to integrate semantic information and collaborative signals during training process. TIGER (Rajput et al., 2023), LC-Rec (Zheng et al., 2024), LETTER (Wang et al., 2024) and TokenRec (Qu et al., 2024) utilize different tokenization strategies to generate high-quality codebooks. Furthermore, LC-Rec, LETTER, SIIT (Chen et al., 2024) and GFlowGR (Wang et al., 2025) integrate tokenization with fine-tuning, enabling the LLMs to better adapt to recommendation-specific tasks. However, the above-mentioned methods only model the inter-layer relationships of codebook in Euclidean space, which fails to capture their intrinsic tree-like hierarchy.

## 6. Conclusion

In this paper, we propose a hyperbolic RQ-VAE enhanced generative recommendation model, namely HG-Rec. Specifically, HG-Rec includes two important components, i.e. hyperbolic RQ-VAE and differential-length codebook strategy. Hyperbolic RQ-VAE adopts hyperbolic residual quantization to capture the hierarchical relationships across codebook layers, and optimizes all residual updates in the tangent space to ensure the stable optimization. Differential-length codebook strategy introduces the growth rate of hyperbolic space to align with the tree-like hierarchy and compress the codebook size. Furthermore, we reveal three intrinsic properties of HG-Rec and extensive experiments demonstrate the effectiveness of our proposed HG-Rec.

## 7. Limitations and Future Work

**Limitations**: One potential limitation of our proposed HG-Rec is that the codebook size for each layer needs to be manually specified for each dataset. Furthermore, the size of each codebook is predefined rather than adaptively learned from data, leading to suboptimal allocation of representational resources. Moreover, we incorporated hyperbolic RQ-VAE to generate a codebook with hierarchical characteristics. Although this does not introduce the extra time cost of downstream tasks, enabling downstream GR models to understand the coarse-to-fine hierarchical characteristics

remains a challenge. Finally, for real-world deployment of recommendation systems, hyperbolic RQ-VAE adopts Poincaré distance to measure the similarities between codewords, which may lead to the problems of unstable training (e.g., numerical instability near the boundary of the Poincaré ball) and high training costs (e.g. expensive hyperbolic operations). Hence, it may limit the potential for large-scale deployment to some extent.

**Future work**: In future work, we will explore semi-supervised learning techniques (Yin et al., 2025; 2026) to learn better representations of entities, which can enhance the performance of GR models and alleviate the problem of original semantic forgetting. Moreover, we plan to incorporate contrastive learning techniques (Zhang et al., 2025b; Yu et al., 2026) to better align the codebook representations with downstream task objectives. Finally, we will investigate replacing the Poincaré model with the Lorentz model to improve training stability and speed.

## Acknowledgements

This work was supported by Fundamental and Interdisciplinary Disciplines Breakthrough Plan of the Ministry of Education of China (No. JYB2025XDXM118), the "111 Center" (No. B26023), the Natural Science Foundation of China (Grant No. 62176119), the Open Foundation of State Key Laboratory for Novel Software Technology at Nanjing University of P.R. China (No. KFKT2025B18), the Future Network Scientific Research Fund Project (FNSRFP-2021-YB-54), and Qing Lan Project of Jiangsu Province.

## Impact Statement

This paper proposes HG-Rec, a hyperbolic RQ-VAE enhanced generative recommendation with differential-length codebook strategy. HG-Rec includes two important designs, i.e. hyperbolic RQ-VAE and differential-length codebook strategy, which captures the hierarchical relationships across codebook layers and effectively compress the codebook size, respectively. Hence, this work aims to advance the efficient and reasonable construction of codebooks for GR, which can further enhance the performance of GRs. We argue that this work is not directly correlated to certain society or ethical concerns. The significance of this work lies chiefly in its broader implications for recommender systems across diverse domains, including e-commerce and social platforms.

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

# A. Notations

In this section, we present the notations used in this paper in Table 4.

*Table 4.* The notations used in this paper.

| Notation | Explaination |
| --- | --- |
| $U, i$ | the sets of all users, items |
| $u, i$ | user ID, item ID |
| $S_u$ | item sequence of user $u$ |
| $A$ | all item sequences |
| $C, C_u$ | All token sequences, token sequence of user $u$ |
| $t$ | the time step in token sequence |
| $L, \ell \in \{1, \cdots, L\}$ | the number of codebook layers, the $\ell^{th}$ layer of codebook |
| $c_\ell$ | the selected codeword index at the $\ell^{th}$ layer of codebook |
| $C_u^{in}$ | the input token sequence of user $u$ for GR |
| $C_{u,t}^{out}$ | the $t^{th}$ token in output token sequence of user $u$ generated by GR |
| $\hat{i}_{t+1}$ | the predicted next item |
| $s$ | semantic embedding |
| $\mathbf{z}$ | latent semantic embedding |
| $K \in \{K_1, \cdots, K_L\}$ | the number of codewords in codebook layer |
| $\mathcal{C}_\ell$ | the embedding set of the $\ell^{th}$ layer of codebook |
| $e_{\ell,*}$ | the embedding of a codeword with index $*$ at the $\ell^{th}$ layer of codebook |
| $c$ | the curvature of hyperbolic space |
| $n$ | the dimension of hyperbolic space |
| $\mathbb{B}_c^n$ | the $n$-dimensional Poincaré-ball with curvature $c$ |
| $g_c^{\mathbb{B}}$ | the Riemannian metric |
| $I_n, g^E$ | Euclidean metric tensor |
| $d_{\mathbb{B}}(\cdot, \cdot)$ | hyperbolic distance |
| $\oplus_c$ | Möbius addition |
| $\mathcal{T}_\mathbf{o}^c \mathbb{B}_c^n$ | tangent space |
| $\gamma$ | the growth rate of codebook size |
| $N$ | the number of candidate items |

# B. Proof of Theorems

## B.1. Proof of Theorem 3.1

*Assuming that RQ-VAE employs $L$ layers of residual quantization, with each layer having a codebook of size $K$. The residual quantization process induces a hierarchical structure on the latent space $\mathbb{R}^n$, which is graph-isomorphic to a rooted $K$-ary tree of depth $L$.*

We provide a complete proof of **Theorem 3.1**.

### B.1.1. DEFINE THE PROCESS OF TRADITIONAL RESIDUAL QUANTIZATION

Given the input vector $\mathbf{z} \in \mathbb{R}^n$ and codebook $\mathcal{C}_\ell = \{\mathbf{e}_{\ell,1}, \ldots, \mathbf{e}_{\ell,K}\}$, the residual quantization algorithm recursively computes a sequence of codewords $(\mathbf{q}_1(\mathbf{z}), \ldots, \mathbf{q}_L(\mathbf{z}))$ and a sequence of residuals $(\mathbf{r}_0(\mathbf{z}), \ldots, \mathbf{r}_L(\mathbf{z}))$, where $\mathbf{r}_0(\mathbf{z}) := \mathbf{z}$.

For $\ell \in \{1, 2, \ldots, L\}$, we select the nearest neighbor codeword in the $\ell^{th}$ layer codebook as follows,

$$i_\ell^*(\mathbf{z}) := \arg \min_{i \in \{1, \ldots, K\}} \|\mathbf{r}_{\ell-1}(\mathbf{z}) - \mathbf{e}_{\ell,i}\|. \tag{14}$$

Record codeword $\mathbf{q}_\ell(\mathbf{z}) := \mathbf{e}_{\ell, i_\ell^*(\mathbf{z})}$, and update residual $\mathbf{r}_\ell(\mathbf{z}) := \mathbf{r}_{\ell-1}(\mathbf{z}) - \mathbf{q}_\ell(\mathbf{z})$. In addition, the sequences $\{\mathbf{q}_\ell(\mathbf{z})\}_{\ell=1}^L$ and $\{\mathbf{r}_\ell(\mathbf{z})\}_{\ell=1}^L$ are uniquely determined. According to the codeword sequence, we can construct a rooted tree $T = (V, E)$. The vertex set

$$V := \bigcup_{\ell=0}^L \Sigma_\ell, where \ \Sigma_\ell := \mathcal{C}_1 \times \mathcal{C}_2 \times \cdots \times \mathcal{C}_\ell \ and \ \Sigma_0 := \{root\}. \tag{15}$$

The edge set

$$E := \{(s, s') : s \in \Sigma_\ell, s' \in \Sigma_{\ell+1}, s' = s \cdot \mathbf{e}\}, \tag{16}$$

where $s \cdot \mathbf{e}$ represents add the codeword $\mathbf{e}$ at the end of $s$.

Define standard $K$-ary tree $T_{\text{std}} = (V_{\text{std}}, E_{\text{std}})$, where the vertex set is $V_{\text{std}} := \bigcup_{\ell=0}^L \{1, \cdots, K\}^\ell$ and the edge set is $E_{\text{std}} := ((i_1, \ldots, i_\ell), (i_1, \ldots, i_\ell, i_{\ell+1})) : (i_1, \ldots, i_\ell) \in \{1, \cdots, K\}^\ell, i_{\ell+1} \in \{1, \cdots, K\}$.

### B.1.2. The proof of Tree Properties

**Lemma B.1.** *The graph $T = (V, E)$ constructed from residual quantization is a rooted tree of depth $L$.*

*Proof.* We have to proof three properties, i.e. exist a unique root, connectivity, and acyclic.

(1) Exist a unique root. $root$ has in-degree zero, while all other nodes have in-degree one.

*Proof.* $root \in \Sigma_0$ is an empty sequence, therefore $\deg^-(root) = 0$. For $\forall s' \in \Sigma_\ell$, $\ell \geq 1$, let $s' = (\mathbf{e}_{1,i_1}, \ldots, \mathbf{e}_{\ell,i_\ell})$, $s := (\mathbf{e}_{1,i_1}, \ldots, \mathbf{e}_{\ell-1,i_{\ell-1}}) \in \Sigma_{\ell-1}$ and $s' = s \cdot \mathbf{e}_{\ell,i_\ell}$. Hence, $(s, s') \in E$ and this representation is unique (the last element of the sequence determines the unique prefix). Therefore, $\deg^-(s') = 1$. $\square$

(2) Connectivity. For any $s \in V$, there exists a unique path from $root$ to $s$.

*Proof.* We use mathematical induction to prove.

Base case: When $|s| = 0$, we have $s = root$. In this case, the trivial path consisting of the single node.

Inductive step: Assume that the statement holds for all sequences of length strictly less than $\ell$.

Consider an arbitrary sequence $s = (\mathbf{e}_{1,i_1}, \ldots, \mathbf{e}_{\ell,i_\ell}) \in \Sigma_\ell$.

Define its prefix $s' := (\mathbf{e}_{1,i_1}, \ldots, \mathbf{e}_{\ell-1,i_{\ell-1}}) \in \Sigma_{\ell-1}$. By the induction hypothesis, there exists a path $root = v_0, v_1, \ldots, v_{\ell-1} = s'$ from the root to $s'$. Since $s = s' \cdot \mathbf{e}_{\ell,i_\ell}$ and the definition of edge, we have $(s', s) \in E$. Therefore, the path can be extended to $root = v_0, v_1, \ldots, v_{\ell-1} = s', v_\ell = s$, which connects $root$ to $s$. $\square$

(3) Acyclic. There is no cycles.

*Proof.* Define a rank function $\text{rank}(s) := |s|$. For any edge, if $(s, s') \in E$, we have $\text{rank}(s') = \text{rank}(s) + 1$. Moreover, any directed path $v_0 \rightarrow v_1 \rightarrow \cdots \rightarrow v_k$ satisfies $\text{rank}(v_0) < \text{rank}(v_1) < \cdots < \text{rank}(v_k)$. Hence, no cycle can exist. $\square$

$\square$

### B.1.3. Graph Isomorphism to a Standard $K$-ary Tree

**Lemma B.2.** *The mapping $\Psi : T \rightarrow T_{\text{std}}$ is a graph-isomorphism.*

*Proof.* We need to prove it from the following four aspects:

(1) Injective mapping. If $s_1 \neq s_2$, then $\Psi(s_1) \neq \Psi(s_2)$.

*Proof.* We use proof by contradiction. Suppose $\Psi(s_1) = \Psi(s_2)$ but $s_1 \neq s_2$. Let $s_1 = (i_1, i_2, \ldots, i_\ell)$ and $s_2 = (j_1, j_2, \ldots, j_\ell)$. Since $\Psi(s_1) = \Psi(s_2)$, by the definition of $\Psi$, we have:

$$(i_1, i_2, \ldots, i_\ell) = (j_1, j_2, \ldots, j_\ell). \tag{17}$$

Therefore,

$$i_k = j_k, \tag{18}$$

which implies $s_1 = s_2$, contradicting our assumption that $s_1 \neq s_2$. $\qquad\square$

(2) Surjective mapping. For any $t \in V_{\text{std}}$, there exists $s \in V$ such that $\Psi(s) = t$.

*Proof.* We use mathematical induction to prove.

Base case: $t = root$ (length 0). Take $s = root \in V$. Then $\Psi(\epsilon) = root = t$.

Inductive step: Assume that the statement holds for all sequences of length strictly less than $\ell$.

Let $t = (i_1, i_2, \ldots, i_\ell)$, and the corresponding codeword index sequence is $s := (i_1, i_2, \ldots, i_\ell)$. We need to show $s \in \Sigma_\ell$ (i.e. each $i_k \in [K]$) to verify $s \in V$. Since $i_k \in \{1, \cdots K\}$ and $\mathcal{C}_k = \{\mathbf{e}_{k,1}, \ldots, \mathbf{e}_{k,K}\}$, $\mathbf{e}_{k,i_k} \in \mathcal{C}_k$. Hence, $s \in \Sigma_\ell \subseteq V$ and $\Psi(s) = \Psi((i_1, i_2, \ldots, i_\ell)) = (i_1, i_2, \ldots, i_\ell) = t$. $\qquad\square$

(3) Preservation of Edges (Forward Direction). If $(s, s') \in E$, then $(\Psi(s), \Psi(s')) \in E_{\text{std}}$.

*Proof.* By the definition of the edge set $E$, if $(s, s') \in E$, then there exists some $\ell \in \{0, \ldots, L-1\}$ and a codeword $\mathbf{e}_{\ell+1, i_{\ell+1}} \in \mathcal{C}_{\ell+1}$ such that

$$s \in \Sigma_\ell, \quad s' \in \Sigma_{\ell+1}, \quad s' = s \cdot \mathbf{e}_{\ell+1, i_{\ell+1}}. \tag{19}$$

and $s' = (\mathbf{e}_{1,i_1}, \mathbf{e}_{2,i_2}, \ldots, \mathbf{e}_{\ell,i_\ell}, \mathbf{e}_{\ell+1,i_{\ell+1}})$. By the definition of the mapping $\Psi$, we have $\Psi(s) = (i_1, i_2, \ldots, i_\ell)$, $\Psi(s') = (i_1, i_2, \ldots, i_\ell, i_{\ell+1})$. According to the definition of the edge set $E_{\text{std}}$, $\big((i_1, \ldots, i_\ell), (i_1, \ldots, i_\ell, i_{\ell+1})\big) \in E_{\text{std}}$. Therefore, $(\Psi(s), \Psi(s')) \in E_{\text{std}}$. $\qquad\square$

(4) Preservation of Edges (Backward Direction). If $(\Psi(s), \Psi(s')) \in E_{\text{std}}$, then $(s, s') \in E$.

*Proof.* This follows directly from the surjectivity of $\Psi$ and the construction of $E$. $\qquad\square$

Above all, the residual quantization process induces a hierarchical structure on the latent space $\mathbb{R}^n$, which is graph-isomorphic to a rooted $K$-ary tree of depth $L$ $\qquad\square$

## B.2. Proof of Theorem 3.2

*The exponential map function* $\exp_{\mathbf{o}}^{(c)} : \mathcal{T}_{o}^c \mathbb{B}_c^n \to \mathbb{B}_c^n$ *is well-defined.* Theorem 3.2 is equivalent to "For any $\mathbf{v} \in \mathbb{R}^n$, the exponential map at the origin satisfies $\exp_{\mathbf{o}}^{(c)}(\mathbf{v}) \in \mathbb{B}_c^n$. "

*Proof.* Assuming that $\mathbf{x} = \exp_{\mathbf{o}}^{(c)}(\mathbf{v})$,

$$\|\mathbf{x}\| = \left\| \tanh\left(\sqrt{c}\|\mathbf{v}\|\right) \frac{\mathbf{v}}{\sqrt{c}\|\mathbf{v}\|} \right\|. \tag{20}$$

Hence,

$$c\|\mathbf{x}\|^2 = c \cdot \frac{1}{c} \tanh^2\left(\sqrt{c}\|\mathbf{v}\|\right) = \tanh^2\left(\sqrt{c}\|\mathbf{v}\|\right), \tag{21}$$

where $\tanh(t) \in (-1, 1)$. For all $t \in \mathbb{R}$, $c\|\mathbf{x}\|^2 = \tanh^2\left(\sqrt{c}\|\mathbf{v}\|\right) < 1$ holds.

When $\mathbf{v} = \mathbf{0}$, $\exp_{\mathbf{o}}^{(c)}(\mathbf{0}) = \mathbf{0} \in \mathbb{B}_c^n$.

When $\|\mathbf{v}\| \to \infty$, $\|\mathbf{x}\| \to \frac{1}{\sqrt{c}}$ and $\|\mathbf{x}\|$ constant less than $\frac{1}{\sqrt{c}}$.

Hence, for any $\mathbf{v} \in \mathbb{R}^n$, the exponential map makes all points fall within Poincaré ball. $\qquad\square$

### B.3. Proof of Theorem 3.3

*The logarithmic map function* $\log_o^c : \mathbb{B}_c^n \to \mathcal{T}_o^c \mathbb{B}_c^n$ *is well-defined.* Theorem 4.2 is equivalent to "For any $\mathbf{x} \in \mathbb{B}_c^n$, logarithmic map function at the origin satisfies $\log_{\mathbf{o}}^{(c)}(\mathbf{x}) \in \mathbb{R}^n$."

*Proof.* For $\mathbf{x} \in \mathbb{B}_c^n$, $c\|\mathbf{x}\|^2 < 1$ and $\sqrt{c}\|\mathbf{x}\| < 1$. Hence, $\sqrt{c}\|\mathbf{x}\| \in [0, 1)$. The logarithmic map function is defined as

$$\mathbf{v} = \log_{\mathbf{o}}^{(c)}(\mathbf{x}) = \frac{1}{\sqrt{c}}\text{arctanh}(\sqrt{c}\|\mathbf{x}\|)\frac{\mathbf{x}}{\|\mathbf{x}\|}. \tag{22}$$

When $\mathbf{x} = \mathbf{0}$, $\lim_{\mathbf{x} \to \mathbf{0}} \log_{\mathbf{o}}^{(c)}(\mathbf{x}) = \lim_{\|\mathbf{x}\| \to 0} \frac{\text{arctanh}(\sqrt{c}\|\mathbf{x}\|)}{\sqrt{c}\|\mathbf{x}\|}\mathbf{x} = \mathbf{0}$

When $\|\mathbf{x}\| \to 0$, $\text{arctanh}(\sqrt{c}\|\mathbf{x}\|) \approx \sqrt{c}\|\mathbf{x}\|$ and $\mathbf{v} \to \mathbf{x}$.

When $\|\mathbf{x}\| \to \frac{1}{\sqrt{c}}$, $\text{arctanh}(\sqrt{c}\|\mathbf{x}\|) \to +\infty$ and $\|\mathbf{v}\| \to +\infty$.

Hence, for any $\mathbf{x} \in \mathbb{B}_c^n$, logarithmic map function at the origin satisfies $\log_{\mathbf{o}}^{(c)}(\mathbf{x}) \in \mathbb{R}^n$. $\qquad\square$

## C. Dataset

Beauty and Instruments are widely used real-world datasets collected from Amazon review, where Beauty contains users' purchase records of cosmetic products and Instruments contains the user interactions with various musical instruments. Different from Amazon datasets, Yelp records user–business interactions collected from the Yelp platform. To investigate the effectiveness of our proposed HG-Rec under different user interaction scenarios, We choose Beauty, Instruments and Yelp to conduct experiments. The statistics of three datasets are presented in Table 5.

*Table 5.* Statistics of three datasets. "**AVg.** *len*" is the average length of item sequences.

| Datasets | #Users | #Items | #Interactions | Sparsity | AVg. *len* |
|:---:|:---:|:---:|:---:|:---:|:---:|
| **Beauty** | 22,363 | 12,101 | 198,502 | 99.926% | 8.87 |
| **Instruments** | 24,772 | 9,922 | 206,153 | 99.916% | 8.32 |
| **Yelp** | 30,431 | 20,033 | 316,354 | 99.948% | 10.40 |

For all datasets, we filter out the users and items that have less than 5 interactions, and utilize the *leave-one-out* strategy (Kang & McAuley, 2018; Rajput et al., 2023; Zhang et al., 2025b) to build training, testing and validation data. Moreover, we truncate item sequences based on the maximum length of 20 for all GR models, and the maximum length of 50 for all traditional sequential recommendation methods, which is consistent with the original papers.

## D. Baselines

We select the following competitive methods as baselines:

### D.1. Traditional sequential recommendation methods

**Caser** (Tang & Wang, 2018) provides a unified and flexible network structure for capturing both general preferences and sequential patterns via horizontal and vertical convolution operations.

**HGN** (Ma et al., 2019) adopts a hierarchical gating network to control what item latent features and which relevant item can be passed to the downstream layers. Moreover, to explicitly capture the item-item relations, HGN utilizes item-item product module to capture sequential patterns.

**SASRec** (Kang & McAuley, 2018) is a unidirectional self-attention model, which captures user's dynamic interests via a self-attention module.

**Bert4Rec** (Sun et al., 2019) represents each item using its unique item identifier. Unlike SASRec, it encodes sequences of item identifiers with a bidirectional Transformer encoder.

## D.2. GR with ID-based tokenization (Hua et al., 2023)

**P5-RID** indicates each item with a random number as the item identifier, which is tokenized into a token sequence based on SentencePiece tokenizer.

**P5-TID** follows a similar design to P5-RID, but replaces random item identifiers with item titles, which are directly tokenized into text tokens by SentencePiece.

**P5-IID** introduces an independent out-of-vocabulary extra token that needs to be learned for each item.

**P5-SemID** utilizes item metadata to construct semantic identifiers for items. These identifiers capture high-level semantic information derived from item content, enabling semantically similar items to share related discrete representations.

**P5-CID** employs spectral clustering based on Spectral Matrix Factorization to generate item indices, which effectively captures the essence of collaborative filtering.

## D.3. GR with context aware-based tokenization

**ActionPiece** (Hou et al., 2025) is the first context-aware tokenization method in recent work for GR. In order to address the lack of context-awareness, ActionPiece constructs its vocabulary by merging feature patterns into new tokens based on their co-occurrence frequencies, both within individual sets and across adjacent sets. Moreover, ActionPiece utilizes set permutation regularization to segment a single action sequence into multiple token sequences with the same semantics, which can enhance the training and inference process.

## D.4. GR with codebook-based tokenization

**TIGER** (Rajput et al., 2023) employs RQ-VAE to discretize item embeddings into semantic tokens, which captures hierarchical and fine-grained semantic information while maintaining high representational capacity. Based on resulting semantic tokens, TIGER trains a Transformer encoder-decoder module to autoregressively generate the next token sequences.

**LC-Rec** (Zheng et al., 2024) adopts the Sinkhorn–Knopp algorithm to enhance the traditional RQ-VAE and incorporates a tuning task to inject collaborative semantics into LLMs.

**LETTER** (Wang et al., 2024) further extends TIGER by injecting collaborative information and diversity-oriented constraints into RQ-VAE.

# E. The pseudo-code of HG-Rec

In this section, we present the pseudo-code of two important components of HG-Rec, i.e.hyperbolic RQ-VAE and differential-length codebook strategy in Algorithm 1 and Algorithm 2, respectively.

---

**Algorithm 1** Pseudo-code of hyperbolic RQ-VAE

---

1: **Input:** latent semantic embedding $z$, codebook $C_\ell = \{e_{\ell,1}, \ldots, e_{\ell,K_\ell}\}$, number of layers $L$
2: **Output:** selected codewords index sequence $c_1, \ldots, c_L$, quantized embedding $\hat{z}$
3: Initialize residual at layer 0 in tangent space: $r_0 \leftarrow z$
4: **for** $\ell = 1$ **to** $L$ **do**
5: $\quad r_{\ell-1}^H \leftarrow \exp_o^{(c)}(r_{\ell-1})$, $e_{\ell,i}^H \leftarrow \exp_o^c(e_{\ell,i})$ # Map to hyperbolic space
6: $\quad c_\ell \leftarrow \arg\min_i d_B(r_{\ell-1}^H, e_{\ell,i}^H)$ # Codeword selection via hyperbolic distance
7: $\quad r_{\ell-1} \leftarrow \log_o^c(r_{\ell-1})$, $e_{\ell,i} \leftarrow \log_o^c(e_{\ell,i})$ # Map to tangent space
8: $\quad r_\ell \leftarrow r_{\ell-1} - e_{\ell,c_\ell}$ # Residual update in tangent space
9: **end for**
10: $\hat{z} \leftarrow \sum_{\ell=1}^{L} e_{\ell,c_\ell}$ # Reconstruct quantized embedding
11: **Return** $c_1, \ldots, c_L, \hat{z}$

---

---

**Algorithm 2** Pseudo-code of Differential-length Codebook Strategy

---

1: **Input:** number of layers $L$, first layer codebook size $K_1$, growth rate $\gamma$
2: **Output:** list of codebook sizes $K_1, \ldots, K_L$
3: Initialize first layer size: $K_1 \leftarrow K_1$
4: **for** $\ell = 2$ **to** $L$ **do**
5: $\quad K_\ell \leftarrow K_{\ell-1} \cdot \gamma$ # Exponential growth of codebook sizes
6: **end for**
7: **Return** $K_1, \ldots, K_L$

---

# F. Implementation details

**Baselines**: The experimental results of Caser, HGN, SASRec, Bert4Rec, P5-RID, P5-IID, P5-TID, P5-SemID, P5-CID, TIGER, LC-Rec and Letter are directly token from the existing works. For other results, we carefully implement the baselines and fine-tune all the hyper-parameters of those baselines with the grid search. For GRs, we implement them via HuggingFace Transformers and PyTorch.

**HG-Rec**: For the process of item tokenization, we adopt hyperbolic RQ-VAE to generate codebook, which is illustrated in Section 3.1. The curvature $c$ of hyperbolic space is set to 1. Since the transformation of mathematics space, we need to tune the loss coefficient $\beta$ within { 0.1, 0.15, 0.2, 0.25, 0.3, 0.5, 1.0 }. Following the setup in TIGER, we utilize sentence-t5-base (Ni et al., 2022) to map the content information (e.g., title, description, and category) into semantic representation. The codebook size adopts differential-length codebook strategy in Section 3.2, i.e. following a pyramidal structure with three layers. For the GR, sentence-t5-base is used as the backbone architecture. The configuration includes a hidden size of 128, a feed-forward inner dimension of 1024, 6 attention heads each with size 64, and ReLU as the activation function. Both the encoder and decoder are constructed with 4 layers. Training is conducted on a single 16-core Intel Xeon Gold 5218 CPU with 128 GB of memory and one RTX 8000 GPU, using a batch size of 256 for 200 epochs. We choose AdamW as optimizer to update the all parameters and the learning rate is tuned within { 0.01, 0.001, 0.0001 }. For model inference, we use beam search with a beam size of 20. The best setting of HG-Rec is presented in Table 6.

*Table 6.* The hyperparameter settings of HG-Rec.

| Stage | Hyperparameter | Beauty | Instruments | Yelp |
|:---:|:---:|:---:|:---:|:---:|
| Hyperbolic RQ-VAE | epoch | 1,000 | 1,000 | 1,000 |
| | learning_rate | 0.001 | 0.001 | 0.001 |
| | weight decay | 0 | 0 | 0 |
| | optimizer | AdamW | AdamW | AdamW |
| | beta | 1.0 | 0.5 | 0.25 |
| | $c$ | 1.0 | 1.0 | 1.0 |
| | codebook_size | [64,128,256] | [64,128,256] | [64,128,256] |
| | layers | [512,256,128,64] | [512,256,128,64] | [512,256,128,64] |
| Generative Recommendation | epoch | 200 | 200 | 200 |
| | learning_rate | 0.0001 | 0.0001 | 0.0001 |
| | drop_rate | 0.1 | 0.1 | 0.1 |
| | beam_size | 20 | 20 | 20 |
| | num_layer | 4 | 4 | 4 |
| | d_model | 128 | 128 | 128 |
| | d_ff | 1024 | 1024 | 1024 |
| | num_heads | 6 | 6 | 6 |
| | d_kv | 64 | 64 | 64 |
| | optimizer | AdamW | AdamW | AdamW |
| | batch_size | 256 | 256 | 256 |
| | early_stop | 20 | 20 | 20 |

# G. Parameter sensitivity analysis

## G.1. The impact of codebook size $K_\ell$

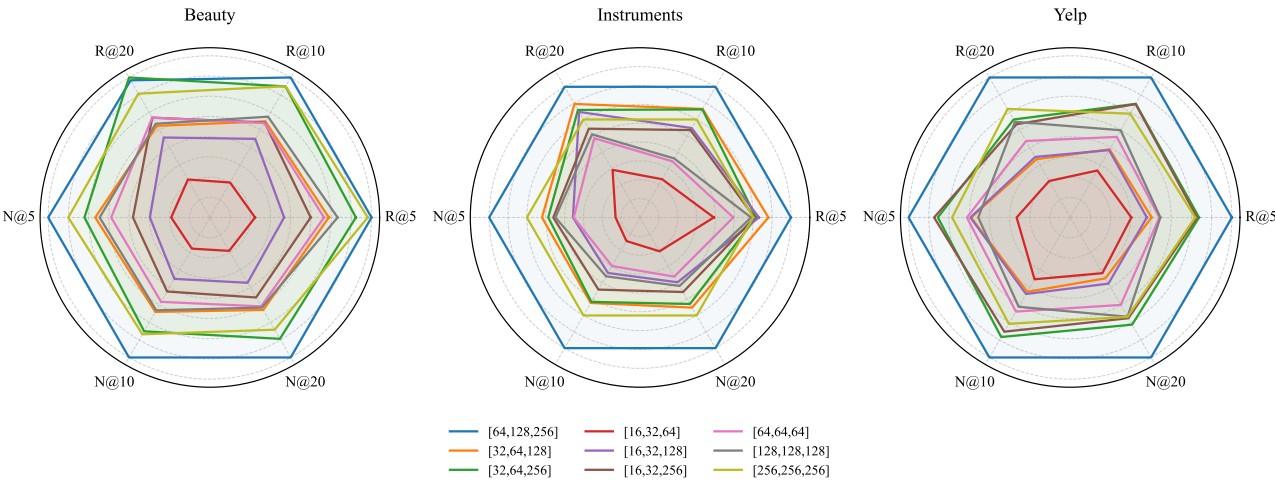

*Figure 7.* The performance comparison of different codebook sizes.

$K_\ell$ is used to control the size of the $\ell^{th}$ codebook. When the growth rate $\gamma$ is set to 1, the differential-length codebook strategy degenerates into the traditional codebook strategy, where the codebook sizes depend on the first codebook size $K_1$, i.e. the codebook sizes are typically set to [64,64,64], [128,128,128] and [256,256,256]. Moreover, for the differential-length codebook strategy, we select the first-layer codebook size $K_1 \in \{16, 32, 64\}$ and tune the growth rate $\gamma \in \{2, 4\}$, i.e. the codebook size are [16,32,64], [16,32,128], [16,32,256], [32,64,128], [32,64,256], [64,128,256], Figure 7 describes the sensitivity of recommendation performance to different codebook size on three datasets. We can draw the following observations:

- On all datasets, larger sizes of codebook generally leads to performance improvements in terms of $Recall@N$ and $NDCG@N$. In particular, the differential-length codebook size [64,128,256] consistently achieves the strongest overall performance, indicating that allocating larger codebook capacities to deeper quantization layers can effectively construct hierarchical codebook.

- The configurations with small codebook, such as [16,32,64], exhibits significantly degraded performance and consistently worse than others under all metrics. This indicates that limited codebook capacity is insufficient to provide adequate discrete representation capability, thereby restricting the discrimination of recommendation models.

- The differential-length codebook strategy generally outperforms traditional codebook strategy in term of performance, which represents that merely increasing the overall capacity without considering hierarchical allocation unable to fully leverage the advantages of hyperbolic residual quantization. In summary, the above results verify the effectiveness of our proposed differential-length codebook strategy.

## G.2. The impact of the loss coefficient $\beta$

Since HG-Rec implements hyperbolic RQ-VAE in hyperbolic space, the original empirical setting (i.e., $\beta = 0.25$) may no longer be optimal. Therefore, we conduct a parameter sensitivity analysis of $\beta$ to investigate its impact on model performance and to identify a suitable operating range for HG-Rec. $\beta$ is tuned within $\{0.1, 0.15, 0.2, 0.25, 0.3, 0.5, 1.0\}$, codebook size is set to [64,128,256], and the experimental results are shown in Figure 8. The performance of HG-Rec increases at the beginning, and gradually reaches its peak when $\beta = 1.0$ on Beauty, 0.5 on Instruments, and 0.25 on Yelp. Then, the performance of HG-Rec begins to degrade. The above results demonstrate that $\beta$ plays a critical role in adjusting the encoder commitment behavior in hyperbolic space. In most cases, both insufficient and excessive $\beta$ lead to performance degradation, while an appropriate intermediate $\beta$ enables HG-Rec to fully exploit hyperbolic codebook representations.

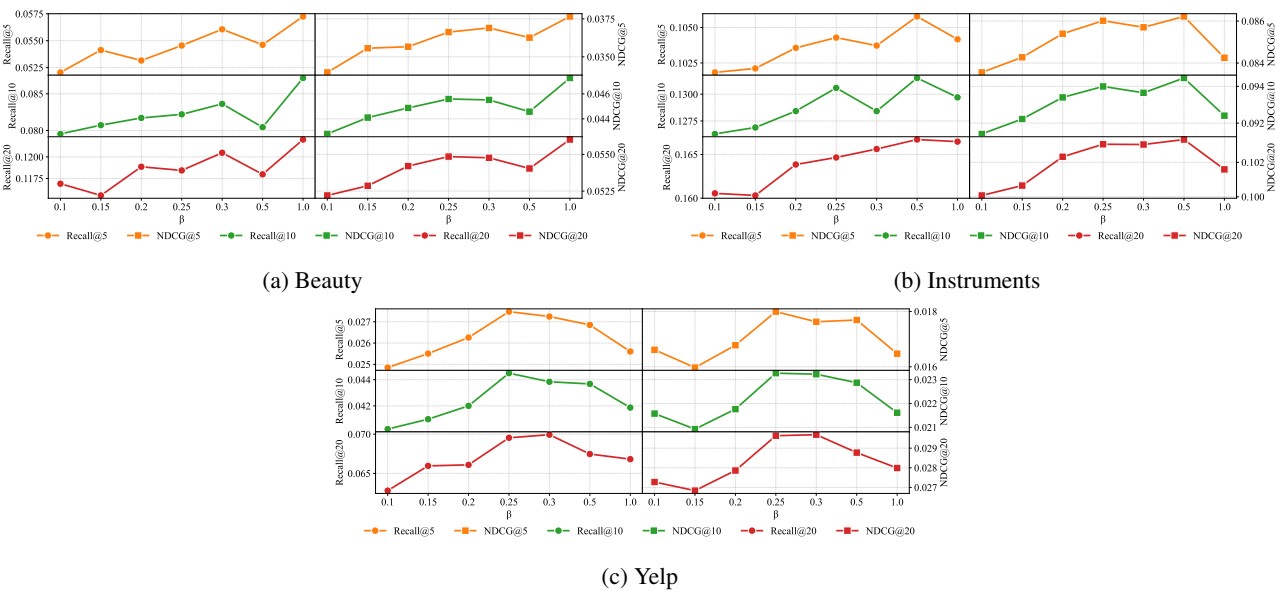

(a) Beauty  (b) Instruments

(c) Yelp

*Figure 8.* Sensitivity analysis of $\beta$ on three datasets.

### G.3. The impact of the codebook utilization

During the recommendation stage, the model only retrieves the codebook three times to obtain candidate items, which reducing the search space and improving efficiency in candidate generation. Our proposed HG-Rec is able to generate a uniformly used codebook via hyperbolic RQ-VAE, and TIGER utilizes Vanilla RQ-VAE to obtain ununiform used codebook. Moreover, the codebook with a lower collision rate usually means that the utilization of codebook is more uniform. To clearly explain how codebook utilization affects the performance of downstream recommendations, we conduct a group of experiments and the detail results are presented in Tables 7 and 8. Specifically, we generate the codebooks with collision rates of around 80%, 60%, 40%, 20% during the training of Hyperbolic RQ-VAE and Vanilla RQ-VAE, respectively.

*Table 7.* The performance of downstream GR under different codebook utilization on HG-Rec.

| Collision Rate | **Beauty** | | | | **Instruments** | | | | **Yelp** | | | |
|---|---|---|---|---|---|---|---|---|---|---|---|---|
| | $R@5$ | $R@10$ | $N@5$ | $N@10$ | $R@5$ | $R@10$ | $N@5$ | $N@10$ | $R@5$ | $R@10$ | $N@5$ | $N@10$ |
| $\approx 80\%$ | 0.0417 | 0.0593 | 0.0269 | 0.0333 | 0.0867 | 0.1049 | 0.0731 | 0.0789 | 0.0209 | 0.0357 | 0.0129 | 0.0178 |
| $\approx 60\%$ | 0.0429 | 0.0630 | 0.0284 | 0.0348 | 0.0913 | 0.1118 | 0.0752 | 0.0818 | 0.0228 | 0.0362 | 0.0143 | 0.0187 |
| $\approx 40\%$ | 0.0460 | 0.0677 | 0.0304 | 0.0374 | 0.0936 | 0.1137 | 0.0783 | 0.0849 | 0.0243 | 0.0402 | 0.0159 | 0.0209 |
| $\approx 20\%$ | 0.0523 | 0.0775 | 0.0347 | 0.0428 | 0.0956 | 0.1162 | 0.0807 | 0.0874 | 0.0261 | 0.0436 | 0.0169 | 0.0225 |
| Best | 0.0572 | 0.0872 | 0.0377 | 0.0473 | 0.1058 | 0.1315 | 0.0862 | 0.0945 | 0.0275 | 0.0445 | 0.0180 | 0.0233 |

We can draw the following conclusion:

- Under the condition of optimal collision rate, HG-Rec is superior to TIGER. The main reason is that Hyperbolic RQ-VAE is able to learn continuous embeddings with discriminative and hierarchical characteristics. The above representations better capture the underlying structure of item relationships, enabling more effective separation of semantically similar items.

- The performance of HG-Rec consistently outperforms TIGER under different parameter settings, which represents that even in the case of codebook collapse, effective modeling of inter-layer relationships of codebook can still improve recommendation performance. This also demonstrates the effectiveness of our proposed HG-Rec.

- As the collision rate increases, the performance of HG-Rec and TIGER gradually decreases. This is because the gradually increasing collision rate limits the representational capability of the codebook, and only a small number of

*Table 8.* The performance of downstream GR under different codebook utilization on TIGER.

| Collision | Beauty | | | | Instruments | | | | Yelp | | | |
|---|---|---|---|---|---|---|---|---|---|---|---|---|
| Rate | $R@5$ | $R@10$ | $N@5$ | $N@10$ | $R@5$ | $R@10$ | $N@5$ | $N@10$ | $R@5$ | $R@10$ | $N@5$ | $N@10$ |
| $\approx 80\%$ | 0.0400 | 0.0590 | 0.0261 | 0.0322 | 0.0891 | 0.1070 | 0.0714 | 0.0789 | 0.0173 | 0.0309 | 0.0112 | 0.0153 |
| $\approx 60\%$ | 0.0415 | 0.0647 | 0.0271 | 0.0346 | 0.0901 | 0.1099 | 0.0728 | 0.0804 | 0.0197 | 0.0331 | 0.0123 | 0.0165 |
| $\approx 40\%$ | 0.0449 | 0.0696 | 0.0293 | 0.0372 | 0.0924 | 0.1106 | 0.0746 | 0.0821 | 0.0213 | 0.0365 | 0.0139 | 0.0187 |
| $\approx 20\%$ | 0.0482 | 0.0731 | 0.0314 | 0.0394 | 0.0947 | 0.1123 | 0.0768 | 0.0841 | - | - | - | - |
| Best | 0.0502 | 0.0775 | 0.0330 | 0.0418 | 0.0979 | 0.1214 | 0.0811 | 0.0886 | 0.0234 | 0.0384 | 0.0154 | 0.0203 |

codewords are frequently used. Consequently, this leads to increased information loss and reduced discriminability among item representations, ultimately degrading the quality of downstream recommendation.

## H. The comparisons of collision rate between Vanilla RQ-VAE and Hyperbolic RQ-VAE

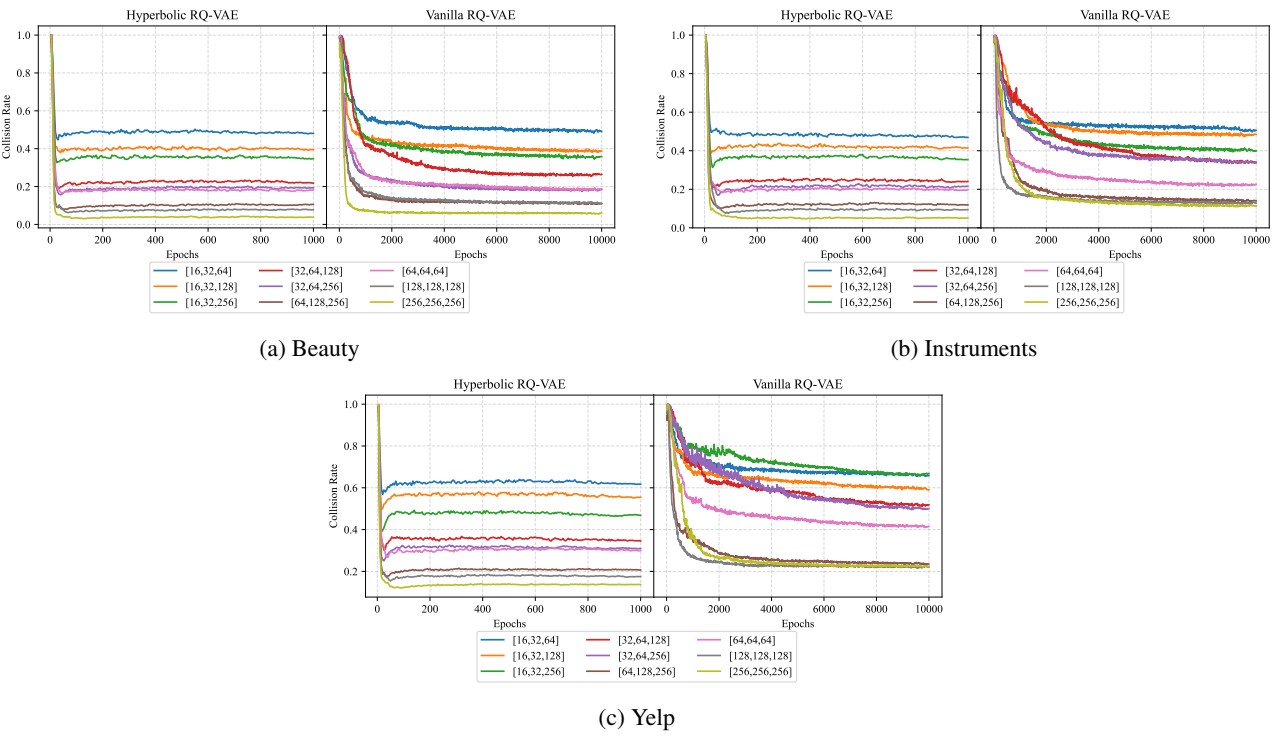

(a) Beauty

(b) Instruments

(c) Yelp

*Figure 9.* The comparisons of collision rates between Vanilla RQ-VAE and Hyperbolic RQ-VAE under different codebook size settings.

In this section, we compare the collision rates between vanilla RQ-VAE and hyperbolic RQ-VAE during training process. Moreover, we evaluate both the traditional codebook strategy and our proposed differential-length codebook strategy when setting the codebook sizes. We employ collision rate to measure whether the codebook is fully utilized. For vanilla RQ-VAE, consistent with previous work (Rajput et al., 2023; Liu et al., 2025; Wei et al., 2025), we set the number of training epochs to 10,000. For hyperbolic RQ-VAE, we observe that the collision rate typically reaches its minimum within 100–300 epoch. Hence, we set the training epochs of hyperbolic RQ-VAE to 1,000. The experimental results are presented in Figure 9 and we can draw the following conclusions:

- For both vanilla RQ-VAE and hyperbolic RQ-VAE, reducing the codebook size consistently leads to an increase in the collision rate, indicating that insufficient codebook capacity limits the diversity of discrete representations. This also affect the quality of downstream GRs (as shown in Figure 7).

- Under the same setting of codebook size, the collision rate of hyperbolic RQ-VAE is lower than that of vanilla RQ-VAE,

which verifies the effectiveness of our proposed hyperbolic RQ-VAE. The main reason is that hyperbolic geometry naturally aligns with the hierarchical structure induced by hyperbolic RQ-VAE, enabling the model to learn more discriminative representations of codewords.

- Compared with vanilla RQ-VAE, hyperbolic RQ-VAE reaches the minimum collision rate much earlier. Although time cost per epoch of hyperbolic RQ-VAE is higher than that of vanilla RQ-VAE (as discussed in Section 4.3.3), hyperbolic RQ-VAE requires fewer training epochs to converge, resulting in efficient overall training process.

# I. The visualization of codebook usage

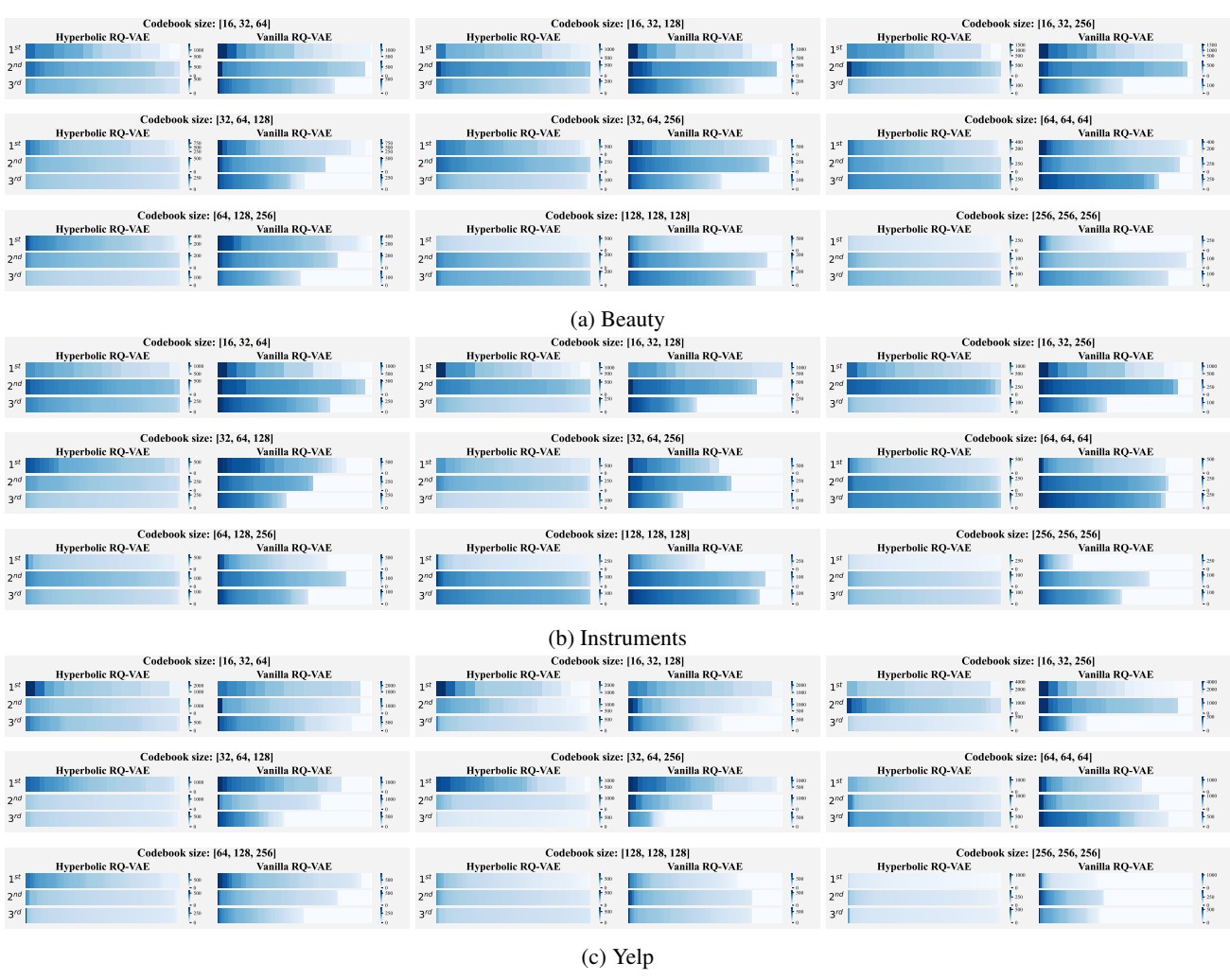

(a) Beauty

(b) Instruments

(c) Yelp

*Figure 10.* The codebook usage of vanilla RQ-VAE and hyperbolic RQ-VAE on three datasets. Darker colors indicate higher usage frequency, while white denotes unused codewords.

In this section, we visualize the codebook usage of vanilla RQ-VAE and hyperbolic RQ-VAE under both traditional codebook strategy and our proposed differential-length codebook strategy. Darker colors indicate higher usage frequency, while white denotes unused codewords. The experimental results are shown in Figure 10 and we can conclude the following observations:

- Hyperbolic RQ-VAE exhibits consistently high codebook utilization, i.e. nearly all codewords are activated. On the contrary, vanilla RQ-VAE suffers from the low codebook usage, where a considerable part of codewords are unused. This indicates that HG-Rec employs hyperbolic RQ-VAE to learn hierarchical representations of items, which effectively improves the usage rate of codebooks.

- Even when the codebook size is reduced to a relatively small scale, vanilla RQ-VAE still fails to fully activate all codewords. The above phenomenon is closely related to feature collapse, as only a limited number of codewords are repeatedly selected. In contrast, the hyperbolic RQ-VAE encourages a more balanced assignment of codewords, because it adopt hyperbolic distances to enhance the discrimination of latent features during quantization.

- The balanced codebook utilization of Hyperbolic RQ-VAE further contributes to lower collision rates and more discriminative discrete representations, which ultimately benefits downstream GR performance.

## J. The visualization of the embeddings of codewords

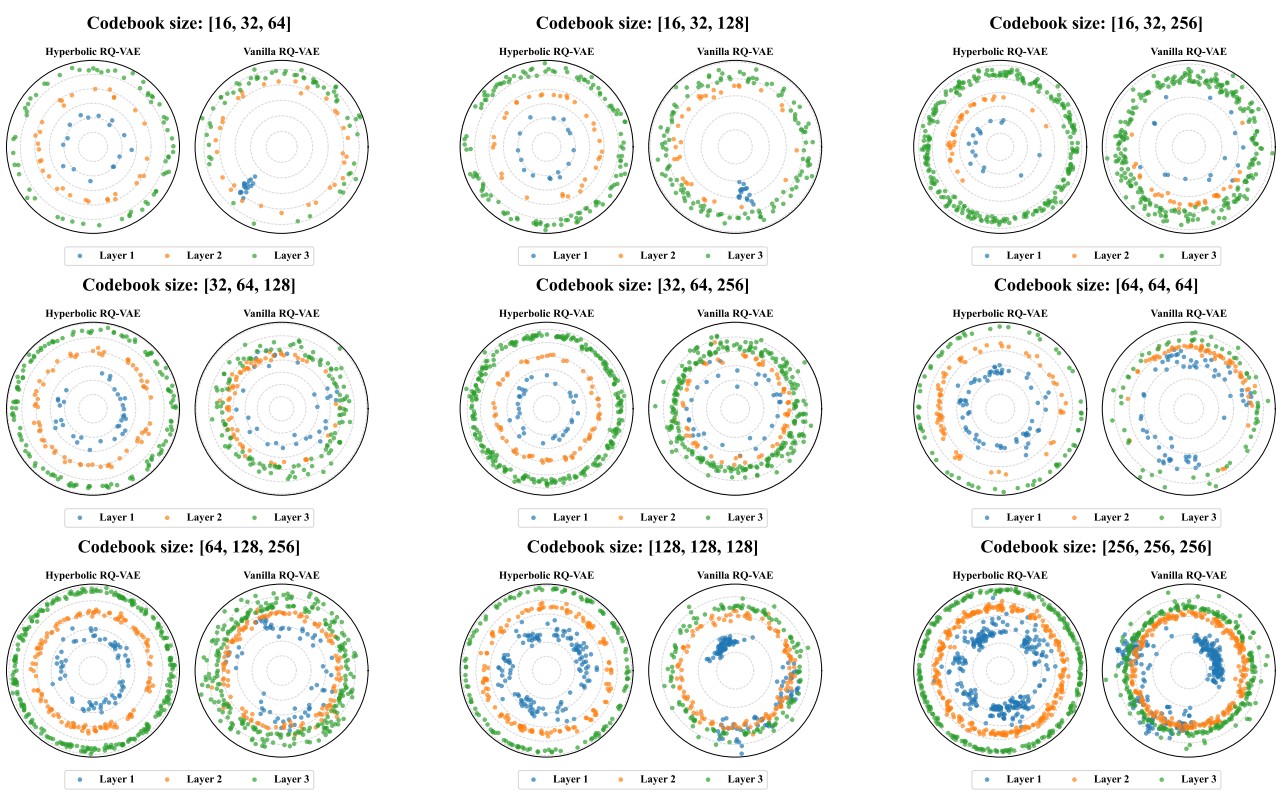

*Figure 11.* The visualizations on Beauty.

In this section, we visualize the codewords' embeddings of vanilla RQ-VAE and hyperbolic RQ-VAE under both traditional codebook strategy and our proposed differential-length codebook strategy. specifically, all codewords embeddings are mapped into a hyperbolic space, where their positions are calculated by hyperbolic distances. The experimental results on three datasets are presented in Figure 11, 12 and 13. We can draw the following observations:

- On all three datasets, the codeword embeddings learned by both vanilla RQ-VAE and hyperbolic RQ-VAE exhibit hierarchical structures to some extent. The hierarchical structure learned by vanilla RQ-VAE provides experimental evidence supporting the correctness of **Theorem 3.1**. Moreover, hyperbolic RQ-VAE demonstrates a clearer hierarchical organization than vanilla RQ-VAE. This improvement can be attributed to the intrinsic geometric properties of hyperbolic space, which naturally aligns with the hierarchical relationships across codebook layers.

- We find that reducing the codebook size leads to noticeable changes in the distribution of embeddings, representing that the operation of reducing the codebook not only a simple compression of the representation capacity, but also affects the distribution and hierarchical structure of representations.

- In most cases, the embedding distribution of hyperbolic RQ-VAE is more uniform than that of vanilla RQ-VAE, indicating that hyperbolic RQ-VAE is able to capture fine-grained embeddings of items. This observation further explains why the codebook usage rate of hyperbolic RQ-VAE is higher than that of vanilla RQ-VAE.

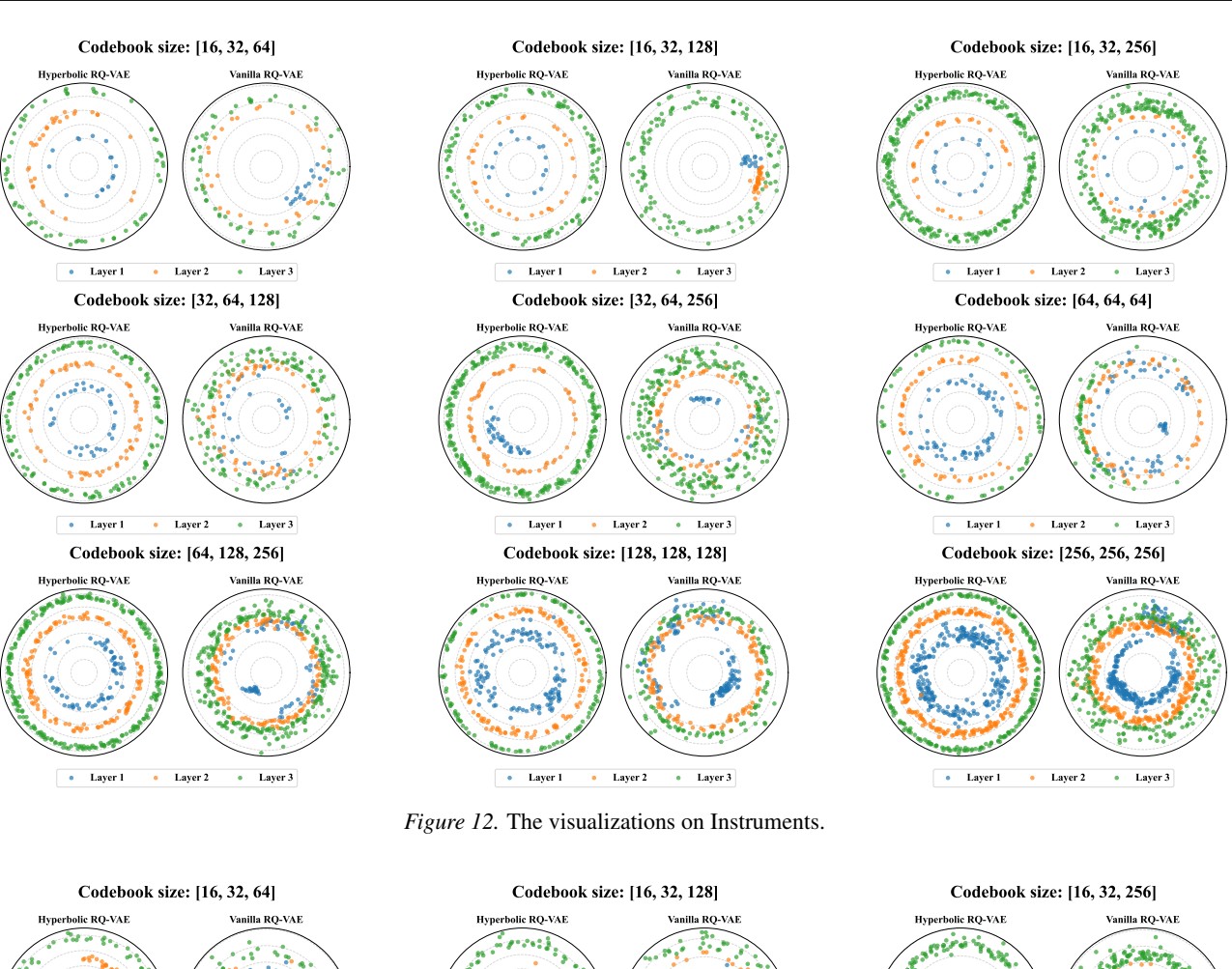

*Figure 12.* The visualizations on Instruments.

*Figure 13.* The visualizations on Yelp.

