# OpenReview forum: "Hyperbolic RQ-VAE enhanced Generative Recommendation with Differential-Length Codebook Strategy"
_ICML.cc/2026/Conference — ICML 2026 regular_

### Official Review · Reviewer_ciCy · 2026-03-08

**Soundness:** 3
**Presentation:** 3
**Significance:** 2
**Originality:** 2
**Overall Recommendation:** 4
**Confidence:** 3

**Summary:**

This paper proposes **HG-Rec**, a framework for Generative Recommendation (GR) that enhances the item tokenization process using **Hyperbolic RQ-VAE**. The authors identify a geometric mismatch in existing codebook-based methods (like TIGER), which model hierarchical residual quantization in Euclidean space despite its intrinsic tree-like structure. To address this, HG-Rec embeds latent representations into a **Poincare ball** and performs residual updates in the **tangent space**. Additionally, motivated by the exponential volume growth of hyperbolic geometry, the paper introduces a **Differential-Length Codebook Strategy** (a pyramidal structure where codebook size grows exponentially with depth). Extensive experiments on three datasets (Beauty, Instruments, Yelp) demonstrate that HG-Rec achieves state-of-the-art performance, significantly reduces codebook collapse, and accelerates training convergence.

**Compliance With Llm Reviewing Policy:**

Affirmed.

**Final Justification:**

The authors' rebuttal has successfully addressed my concerns, therefore I am giving a positive final score.

**Key Questions For Authors:**

1. **Scalability:** How does HG-Rec perform on larger-scale benchmarks such as **ML-20M** or larger Amazon categories with $>10^5$ items? For a corpus of $1$ million items, how should the hyperparameters $L$ (layers) and $K$ (codebook sizes) be optimally configured under the differential-length strategy to balance accuracy and latency?
2. **Inference Latency:** Could you provide a concrete measurement of the inference latency (e.g., in milliseconds per request) during the beam search decoding phase? Specifically, how much overhead do the hyperbolic distance calculations and tangent space mappings introduce compared to a standard Euclidean RQ-VAE?
3. **LLM Compatibility:** Since the current backbone is a small Transformer, have you experimented with using the learned hyperbolic tokens as input to a *real* pre-trained LLM (e.g., Llama-2-7B or T5-Large)? Does the hyperbolic structure of the tokens align well with the pre-trained embedding spaces of these large models?

**Limitations:**

The authors do not explicitly dedicate a section to discussing limitations in the main text.
1. **Scalability:** The primary limitation is the unproven scalability of the Hyperbolic RQ-VAE to massive, industry-scale item catalogs (millions of items). The tree depth required for such scales might introduce significant latency.
2. **Deployment Costs:** The reliance on complex Riemannian geometry operations (exponential/logarithmic maps and Mobius addition) introduces computational bottlenecks that may hinder deployment on edge devices or strictly latency-constrained real-time serving environments compared to simple Euclidean dot-product lookups.

**Strengths And Weaknesses:**

**Strengths:**

1.  **Theoretical Alignment of Geometry and Hierarchy:** The motivation to replace Euclidean embedding spaces with Hyperbolic geometry is theoretically sound. Since residual quantization (RQ) naturally induces a structure similar to a rooted tree, modeling this in a hyperbolic space (Poincare ball) effectively minimizes the distortion of hierarchical relationships compared to flat Euclidean spaces.

2.  **Exceptional Visualization and Empirical Evidence:** The paper provides compelling visual evidence to support its claims. The visualizations of codebook usage (Figure 5) effectively demonstrate that the proposed method solves the "codebook collapse" problem, achieving nearly 100% utilization. Furthermore, the embedding visualizations on the Poincare disk (Figures 11-13) offer a clear, intuitive confirmation that the hyperbolic model successfully disentangles fine-grained semantic hierarchies.

3.  **Practical Efficiency via Differential Strategy:** The proposed **Differential-Length Codebook Strategy** is a significant practical contribution. By allocating capacity according to the "coarse-to-fine" granularity of features (a pyramidal structure where codebook size grows with depth), it avoids parameter redundancy in earlier layers. This strategy is shown to not only align with the exponential capacity of the manifold but also empirically reduce the training epochs required for convergence.

**Weaknesses:**

1.  **Misleading Terminology Regarding "Large Language Models":** The title and abstract heavily emphasize the integration of "Large Language Models (LLMs)." However, a close inspection of the implementation details (Appendix E) reveals that the generative backbone is merely a 4-layer Transformer with a hidden dimension of 128. In the context of modern AI, referring to such a small, scratch-trained architecture as an "LLM" is terminologically misleading and overstates the paper's connection to foundation models.

2.  **Limited Evaluation Scale (Scalability Concerns):** The evaluation is restricted to relatively small datasets (Beauty, Instruments, Yelp) with item counts ranging from 10k to 20k. Industrial recommendation systems typically operate on catalogs with millions of items. It remains unproven whether the proposed differential strategy (e.g., capping the outer layer size at 256) can scale to millions of items without requiring an exponentially deeper codebook depth, which would negatively impact inference latency.

3.  **Computational Overhead of Manifold Operations:** While the total training time is reduced due to faster convergence, the *per-epoch* training time is significantly higher (1.5x to 2x) compared to the Euclidean baseline. This is due to the expensive exponential and logarithmic map operations required for hyperbolic geometry. The paper lacks a critical analysis of how these operations affect **online inference latency** (milliseconds per query), which is a vital metric for real-time systems.

4.  **Suboptimal Presentation and Notation Quality:** The manuscript lacks polish in its visual presentation. A notable example is **Figure 3**, where mathematical symbols are represented using raw text/code formats with underscores (e.g., `<a_1>`, `<b_3>`, `<c_4>`) rather than proper mathematical typesetting (e.g., $a_1$, $b_3$). This inconsistent notation gives the impression of unrendered LaTeX code or raw debug logs, detarcting from the rigor expected in a conference publication.

---

> ### Author Rebuttal · Authors · 2026-03-30
>
> We sincerely thank the valuable feedback provided by Reviewer **ciCy**. We present our point-by-point responses to the comments as follows:
>
> ---
> **W1 & Q3:** The problem of **LLM Compatibility**.
>
> **Author Response:** Based on your valuable feedback, we have revised the relevant statements and added the experiment of changing the downstream GR model to LLM. Specifically, we employ the T5 series model as the backbone of GR and conduct experiments on three datasets, as follows:
>
> Table 1: The impact of LLM backbone
> ||#Parameters|Beauty|Instruments|Yelp|
> |-|-|-|-|-|
> |HG-Rec|0.05B|0.0572|0.1058| 0.0275|
> |+T5-Small|0.22B|**0.0574**|0.1045|**0.0295**|
> |+T5-Base|2.85B|**0.0581**|**0.1064**|**0.0307**|
> |+T5-Large|7.38B|0.0565|0.1044|**0.0317** |
>
> We observe that smaller datasets perform best with **T5-Base**, suggesting that limited interaction data may lead to overfitting when using larger LLMs. In contrast, larger datasets achieve the best performance with **T5-Large**, indicating that models with higher capacity can better capture complex user–item relationships when sufficient data is available. In addition, the experimental results also confirmed that the hierarchical codewords sequences are well aligned with the pre-trained embedding spaces of LLM.
>
> ---
> **W2 & Q1:** The problem of **Scalability**.
>
> **Author Response:** To better evaluate the performance of HG-Rec on large-scale datasets, we selected Games, ML20m, Arts, Cell_Phone for further experiments, and statistics summarized as follows:
>
> Table 2: Statistics
> |Dataset|#user|#item|
> |-|-|-|
> |Games|50,545|16,858|
> |ML20m|138,492|26,743|
> |Arts|197,285| 89,957|
> |Cell_Phone|380,998|111,479|
>
> We choose HG-Rec and TIGER to construct the codebook with [64,128,256], and the specific results are as follows
>
> Table 3: HG-Rec
> |Dataset|Collision Rate|Per(s)|Best(-th)|Total(s)|
> |-|-|-|-|-|
> |Games|0.1007|0.80|220|208|
> |ML20m|0.0439|1.29|190|247|
> |Arts|0.2634|3.49|200|777|
> |Cell_Phone|0.3506|4.14|340|1593|
>
> Table 4: TIGER
> |Dataset|Collision Rate|Per(s)|Best(-th)|Total(s)|
> |-|-|-|-|-|
> |Games|0.1155|0.43|9,960|4,482|
> |ML20m|0.0574|0.57|9,880|5,971|
> |Arts|0.3633|1.86|9,480|18,864|
> |Cell_Phone|0.4041|2.30|9,710|23,545|
>
> When handling 10k items, the collision rate increases significantly. This issue can be mitigated by progressively enlarging the capacity of each layer(i.e. start with the 3rd layer, followed by the 2nd, and finally the 1st), as shown below
>
> Table 5: HG-Rec
> |Dataset|Collision Rate|Per(s)|Best(-th)|Total(s)|
> |-|-|-|-|-|
> |Arts+[64,128,512]|0.2179|3.58|180|702|
> |Arts+[64,256,512]|0.1572|3.77|240|921|
> |Arts+[128,256,512]|0.1032|4.01|260|1073|
> |Cell_Phone+[64,128,512]|0.3101|4.23|280|1239|
> |Cell_Phone+[64,256,512]|0.2443|4.39|210|999|
> |Cell_Phone+[128,256,512]|0.1845|4.45|170|814|
>
> In particular, we do not recommend increasing the number of codebook layers, which implies that the codeword sequences generated from the codebook will grow multiplicatively. In large-scale datasets, this will significantly increase both the difficulty and the computational overhead of downstream generative recommendation (GR) tasks. Moreover, we try our best to conduct a series of experiments on several large-scale datasets to investigate the downstream GR performance, as follows:
>
> Table 6: The performance of HG-Rec on large datasets
> ||TIGER|HG-Rec|
> |-|-|-|
> ||R@5|R@5|
> |Games|0.0414|**0.0473**|
> |ML20M|0.1344|**0.1401**|
>
> ---
> **W3 & Q2**: The problem of **Inference Latency**.
>
> **Author Response:** We measure the computational cost of hyperbolic and Euclidean operations in milliseconds, as shown below:
>
> Table 7: Time comparison
> |Exponential map|Logarithmic map|Hyperbolic distance|Euclidean distance|
> |-|-|-|-|
> |0.57ms|0.44ms|1.36ms|1.05ms|
>
> We find that complex hyperbolic operations introduce additional computational overhead, and the overall time cost is approximately 1.5×–2× that of the Euclidean space, which is consistent with our previous time comparisons. Essentially, we only employ hyperbolic operations during the codebook generation process, while the codeword sequences fed into downstream GR tasks are obtained by retrieving from the pre-generated codebook. Hence, this does not increase the generation time of the next item in the GR models or the online inference latency in real-time systems.
>
> ---
> **W4:** The problem of **typos**.
>
> **Author Response:** We sincerely appreciate your careful and thorough review. We have corrected the errors in Fig.3, added the missing axis ticks in Fig.7 and 11, fixed the overflow issue in Fig.12, and corrected the notation error in line 865, and other minor typos. We will revise the above issues in the final version.
>
> ---
> Missing **Limitations** section.
>
> **Author Response:** We appreciate your detailed feedback. We have carefully considered all the limitations mentioned by the four reviewers and added **Limitations and Future work**. Please kindly refer to our response to **Missing Limitations section** of Reviewer **ww3f** for a detailed explanation.

---

> > ### Author Rebuttal · Reviewer_ciCy · 2026-04-03
> >
> > Thanks for the detailed experiments. I will maintain my positive score.

---

> > > ### Author Response · Authors · 2026-04-04
> > >
> > > We thank reviewer **ciCy** for the dedicated and professional comments. In the final version, we will revise the aforementioned issues, particularly by incorporating experiments that use LLMs as the generative recommendation model, and by elaborating on the **Limitations and Future Work**.

---

### Official Review · Reviewer_bWwG · 2026-03-09

**Soundness:** 3
**Presentation:** 3
**Significance:** 3
**Originality:** 3
**Overall Recommendation:** 4
**Confidence:** 4

**Summary:**

The paper proposes HG-Rec, a generative recommendation framework that introduces hyperbolic residual quantization together with a differential-length codebook design. The main idea is to better match the potentially hierarchical structure of multi-layer codebooks with hyperbolic geometry, instead of modeling all layer relationships in Euclidean space.
The paper’s main contributions are:
1. proposing a hyperbolic RQ-VAE design for multi-layer codebook learning,
2. introducing a layer-wise codebook allocation strategy that assigns different capacities to different quantization levels, and
3. showing through experiments on multiple datasets that the proposed design improves recommendation performance while also leading to better tokenizer-related properties such as lower collision, more balanced code usage, and clearer hierarchical organization.

**Compliance With Llm Reviewing Policy:**

Affirmed.

**Key Questions For Authors:**

1. Could the authors further clarify which observations or analyses mainly support the core assumption that multi-layer codebooks exhibit a hierarchical structure?
2. Since the ablation study already shows that both hyperbolic geometry and allocation adjustment contribute to the performance gains, could the authors further clarify the complementary roles of these two components in the overall method?
3. Could the authors discuss more explicitly why utilization imbalance is expected to affect downstream recommendation performance?

**Strengths And Weaknesses:**

Strengths
1. The motivation is clear and the overall narrative is coherent.
The paper is centered around the idea that multi-layer codebooks may exhibit a hierarchical structure, while Euclidean space may not be the most suitable geometry for modeling such inter-layer relations. From this perspective, introducing hyperbolic geometry is conceptually natural.
2. The method design is relatively clean.
Both the hyperbolic RQ-VAE and the differential-length codebook strategy serve the same core motivation, rather than being a simple combination of multiple independent techniques.
3. The empirical results are solid.
HG-Rec consistently outperforms existing baselines across multiple datasets and evaluation metrics, making the empirical results fairly convincing.
4. The paper includes useful tokenizer-level analyses.
The additional analyses on collision rate, codebook usage, training time, and hierarchy visualization make the paper stronger than one that only compares final recommendation performance.

Weaknesses
1. The evidence for the core mechanism remains somewhat indirect.
The paper shows lower collision rate, more balanced code usage, and better downstream performance, which together support the effectiveness of the proposed method. However, the paper could further clarify whether these gains mainly come from better modeling of hierarchical structure in hyperbolic space, rather than simply from an overall improvement in the quantization process.
2. The justification for the hierarchical structure of multi-layer codebooks could be made clearer.
The paper treats the hierarchical structure of multi-layer codebooks as one of its core motivations. If the authors could more clearly explain which observations or analyses mainly support this assumption, the overall argument would be more complete.
3. The differential-length codebook strategy itself also appears to play an important role.
The ablation study already shows that both hyperbolic geometry and allocation adjustment contribute to the final performance. It would be helpful if the authors could further clarify the complementary roles of these two components in the overall method.
4. The connection between utilization imbalance and downstream recommendation could be explained more clearly.
The overall intuition is reasonable, but the paper would be more complete if this connection were made more explicit. For example, the authors could further discuss whether utilization imbalance mainly affects the effective code capacity or introduces ambiguity at the tokenization level.

---

> ### Author Rebuttal · Authors · 2026-03-30
>
> We sincerely thank the valuable feedback provided by Reviewer **bWwG**. We present our point-by-point responses to the comments as follows:
>
> ---
> **W1:** The problem of **the effectiveness of hyperbolic embeddings**.
>
> **Author Response:** To better investigate whether the performance gains of HG-Rec come from quantization or hierarchical modeling, we conduct a group of experiments on Beauty. Specifically, we first remove the differential-length codebook strategy from HG-Rec, resulting in the variant **HG-Rec/DL**, and then further remove the hyperbolic modeling, yielding **HG-Rec/H**. The results are as follows:
>
> ||R@5|R@5|
> |-|-|-|
> |Codebook size|HG-Rec/DL|HG-Rec/H|
> |[64,64,64]|0.05134|0.04910|
> |[128,128,128]|0.05226|0.04938|
> |[256,256,256]|0.05430|0.05023|
>
> The performance of HG-Rec/DL is significantly higher than that of HG-Rec/H, showing that replacing hyperbolic distance with Euclidean distance under the same quantization framework leads to clear performance degradation. The aforementioned phenomenon can be attributed to the natural geometric alignment between the codebook latent space and hyperbolic space. Hence, the performance improvement of GR comes from hyperbolic embeddings, rather than the residual quantization process.
>
> ---
> **W2 & Q1:** The problem of **hierarchical structure of multi-layer codebooks**.
>
> **Author Response:** Traditional RQ-VAE encodes items into three discrete codewords across three different layers, which inspired us to further explore whether there exists a specific relationship between different layers. Hence, we introduce a case study (e.g., Figure 1) to investigate the relationships among different layers. We found that the process of residual quantization effectively groups items based on their semantic information, and deeper layers correspond to progressively finer semantic refinement. Moreover, this process essentially corresponds to finding an acyclic path that captures semantics from coarse to fine, which is analogous to constructing a tree-like structure. Hence, we proved **the residual quantization process induces a hierarchical structure on the latent space, which is graph-isomorphic to a rooted tree** (e.g., Theorem 3.1). However, embedding a tree structure in Euclidean space leads to structural distortion, and we have proved this theorem in **W1 & Q2** of Reviewer **ww3f**. Therefore, we model the quantization process in hyperbolic space and effectively preserve the hierarchical relationships across layers. In our paper, we demonstrate the hierarchical structure of multi-layer codebooks via case study (e.g., Figure 1) and theoretical proof (e.g., Theorem 3.1). We hope this addresses your concern.
>
> ---
> **W3 & Q2**: The problem of **the complementary roles of two proposed components**.
>
> **Author Response:** We clarify the complementary roles of the two components as follows. The **hyperbolic RQ-VAE** primarily improves the **latent space** of codebook via aligning the hierarchical structure and hyperbolic geometry. This enables the model to learn more discriminative embeddings, thereby generating higher-quality codebooks and effectively reducing the collision rates.  The **differential-length codebook strategy** operates at the aspect of **capacity allocation**, which allows the model to allocate more capacity to fine-grained layers while avoiding redundancy in coarse layers. Therefore, the two proposed components enhance the traditional RQ-VAE from two different perspectives. Especially, there is no conflict between components, as differential-length codebook strategy is based on the growth rate of hyperbolic space volume, which naturally fits hyperbolic space. This clarifies their complementary roles in jointly improving representation quality and allocation efficiency.
>
> ---
> **W4 & Q3**: The problem of **codebook utilization**.
>
> **Author Response:** We thank the reviewer for raising this important question. Similar concerns are also discussed in Reviewer **ww3f**’s **W2 & Q1** and Reviewer **zNoz**’s **Q3**, where we provide additional experiments analyzing the impact of codebook utilization (i.e., collision rate) on downstream GR performance. The corresponding supplementary analysis is as follows:
>
> The uniform codebook utilization typically implies two situations, (1) **Lower collision rate**: Hyperbolic embeddings preserve the hierarchical structure of the codebook with high fidelity, enabling the model to learn more discriminative representations and reducing the likelihood that similar items are mapped to the same codeword. (2) **Codebook Compression**: Due to more uniform codebook utilization, redundancy among codewords is reduced. Together with the exponential growth property of hyperbolic space, this observation motivates us to propose a differential-length codebook strategy. Essentially, compressing the codebook reduces the search space over codewords for downstream GR, thereby alleviating both generation and recommendation difficulty to some extent.

---

> > ### Author Rebuttal · Reviewer_bWwG · 2026-04-04
> >
> > Thanks for the response and I will keep my positive score.

---

> > > ### Author Response · Authors · 2026-04-04
> > >
> > > We thank the reviewer **bWwG** for the constructive feedback and are glad that the concerns have been addressed. In the final version, we will revise the aforementioned issues, with a particular focus on clarifying the complementary roles of the two proposed components and analyzing the impact of codebook utilization on downstream recommendation performance.

---

### Official Review · Reviewer_zNoz · 2026-03-13

**Soundness:** 3
**Presentation:** 3
**Significance:** 2
**Originality:** 2
**Overall Recommendation:** 4
**Confidence:** 4

**Summary:**

The paper proposes HG-Rec, a generative recommendation framework built upon RQ-VAE. The key idea is to embed the residual quantization process into hyperbolic space to better capture hierarchical relationships across codebook layers. This design aims to improve codebook utilization, reduce collisions, and better align with the hierarchical structure of discrete representations. Experiments on several benchmark datasets show improvements over existing generative recommendation methods in both performance and training efficiency.

**Compliance With Llm Reviewing Policy:**

Affirmed.

**Final Justification:**

The rebuttal has addressed my concerns.

**Key Questions For Authors:**

1. How sensitive is the performance to the design of the differential-length codebook (e.g., the pyramid structure and specific size ratios)? Would different configurations significantly affect performance or training stability?

2. What is the computational overhead introduced by hyperbolic space operations compared with standard Euclidean RQ-VAE? Could this become a bottleneck in large-scale recommendation systems?

3. Can the authors providea  more detailed analysis of how hyperbolic geometry improves codebook utilization or reduces collision rates compared with Euclidean representations?

**Limitations:**

No. The paper could benefit from a clearer discussion of practical limitations, such as scalability, computational cost of hyperbolic operations, and potential constraints when applying the model in real-world recommendation systems.

**Strengths And Weaknesses:**

## **Soundness**

### **Strengths:**

The proposed method is technically reasonable and the motivation is relatively clear. Modeling hierarchical structures in hyperbolic space is well aligned with its geometric properties, and applying it to the residual quantization process is a sensible design choice. The paper also provides empirical evidence on multiple datasets, and the evaluation compares the method against several relevant baselines.

### **Weaknesses:**

Some design choices are not fully justified. For example, the specific configuration of the differential-length codebook and its influence on model behavior are not thoroughly analyzed. In addition, the paper mainly relies on empirical results, while deeper theoretical or analytical explanations of why hyperbolic geometry improves quantization hierarchy are limited.

## **Presentation**

### **Strengths:**

Overall, the paper is reasonably organized, with a clear structure covering motivation, method design, and experiments. The main idea of aligning hyperbolic geometry with hierarchical codebooks is understandable, and the experimental section presents results across several datasets.

### **Weaknesses:**

Some parts of the method description could be clearer, particularly the interaction between the hyperbolic embedding and the residual quantization stages.

## **Significance**

### **Strengths:**

The paper targets generative recommendation, which is an increasingly important direction in recommendation systems and large-model-based recommendation frameworks. Improving discrete representation learning and codebook utilization is relevant for scalability and efficiency, and the idea of incorporating non-Euclidean geometry into quantization-based recommendation models could inspire further exploration in this area.

### **Weaknesses:**

The practical impact of the method is somewhat unclear. It would be helpful to better analyze the computational overhead introduced by hyperbolic operations and whether the approach scales well to very large recommendation systems.

## **Originality**

### **Strengths:**

The work combines several existing ideas, including hyperbolic representation learning, residual quantization, and generative recommendation, in a relatively creative way. Applying hyperbolic geometry to model hierarchical relationships in RQ-VAE style architectures is an interesting perspective and provides a new variant of quantization-based generative recommendation models.

### **Weaknesses:**

While the combination is interesting, many individual components (hyperbolic embeddings, RQ-VAE style quantization, hierarchical modeling) are already well studied. The novelty mainly lies in their integration.

---

> ### Author Rebuttal · Authors · 2026-03-30
>
> We greatly appreciate the insightful suggestions provided by Reviewer **zNoz**. We present our point-by-point responses to the comments as follows:
>
> ---
> **W1 & Q1:** The problem of **different codebook size configurations**.
>
> **Author Response:** We would like to clarify that the detailed experiment and analysis of the impact of the codebook sizes have already been provided in **Appendix F.1**. We sincerely apologize for the oversight in omitting the scale in **Fig.7**, which may hinder intuitive interpretation of the results. In the final version, we will add the missing scale and move **Appendix F.1** into the main body. To better address this concern, we provide the performance under different codebook size settings as follows:
>
> ||Beauty|Instruments|Yelp|
> |-|-|-|-|
> |Size|R@5|R@5|R@5|
> |[128,128,128]|0.05226|0.10208|0.02504|
> |[256,256,256]|0.05430|0.10220|0.02623|
> |[32,64,128]|0.05167|0.10304|0.02475|
> |[32,64,256]|0.05350|0.10236|0.02636|
> |[64,128,256]|0.05723|0.10576|0.02748|
>
> ---
> **W1 & W2:** The problem of **the description and analysis of hyperbolic RQ-VAE**.
>
> **Author Response:** In order to make the process of hyperbolic RQ-VAE more clear, we will add the pseudo-code into the final version, as follows:
>
> **Algorithm 1 Hyperbolic RQ-VAE**
>
> **Input:** the latent semantic embedding $z$, codebook $C_{\ell}=\{e_{\ell,1},\cdots,e_{\ell,K_{\ell}}\}$, the layer of codebook $L$
>
> **Output:** the selected codeword index sequence $c_1,\cdots,c_L$, the quantized embedding $\hat{z}$
>
> 1. Initialize the residual at the $0^{th}$ layer in tangent space: $r_0 \leftarrow z$
> 2. **For** $\ell=1$ to $L$ **do**
>    - $r^H_{\ell-1} \leftarrow\exp_o^c(r_{\ell-1}),e^H_{\ell,\*}\leftarrow\exp_o^c(e_{\ell,\*})$ #Map to hyperbolic space
>    - $c_\ell \leftarrow\arg\min_id_B(r^H_{\ell-1},e^H_{\ell,i})$ #Codeword selection via hyperbolic distance
>    - $r_{\ell-1}\leftarrow\log_o^c(r_{\ell-1}^H),e_{\ell,\*}\leftarrow\log_o^c(e_{\ell,\*}^H)$ #Map to tangent space
>    - $r_\ell\leftarrow r_{\ell-1}-e_{\ell,c_\ell}$ #Residual update in tangent space
> 3. **End for**
> 4. $\hat{z} \leftarrow \sum_{\ell=1}^{L}e_{\ell,c_\ell}$ #Reconstruct quantized embedding
>
> **return** $c_1,\cdots,c_L, \hat{z}$
>
> Moreover, we add the following **analysis of hyperbolic RQ-VAE** in Section 3.1,
>
> **Discussion:** The process of residual quantization is essentially inducing a coarse-to-fine tree-like structure in latent space. However, traditional RQ-VAE embeds this structure in Euclidean space, which not only distorts the inherent hierarchy but also leads to inefficient representation. This is because the volume grows of Euclidean space only polynomially with respect to the radius, leading to crowding effects and insufficient separation between codewords at deeper levels. In contrast, our proposed hyperbolic RQ-VAE benefits from the natural geometric alignment between hyperbolic space and tree-like structures, providing low distortion embeddings for entities at different levels. Hence, the model can learn more discriminative representations, thereby achieving more uniform codebook utilization and a lower collision rate.
>
> ---
> **W3 & Q2:** The problem of **computational overhead of HG-Rec**.
>
> **Author Response:** The time complexity of RQ-VAE is $O(LKd)$, while that of hyperbolic RQ-VAE is $O(L\cdot(2Kd+2Kd))$, where $O(2Kd)$ is from hyperbolic distance, $O(2Kd)$ is from hyperbolic mapping function. Moreover, we also provide detailed training time cost in **Section 4.3.3**.
>
> In addition, for the possibility of scaling to very large recommendation systems, we also provide an analysis using LLM as the generator and large datasets. Please kindly refer to our response to **W1 & Q3** and **W2 & Q1** of Reviewer **ciCy** for detailed explanations.
>
> ---
> **Q3:** The problem of **codebook utilization**.
>
> **Author Response:** In the **Discussion** sections of **W1 & W2**, we provide a more detailed analysis of how hyperbolic geometry improves codebook utilization. In addition, we conduct experiments in response to Reviewer **ww3f**’s **W2 & Q1** and Reviewer **bWwG**’s **W4 & Q3** to further investigate the impact of codebook utilization. We hope these will help address your question.
>
> ---
> **W4:** The problem of **novelty**.
>
> **Author Response:** We agree that the individual components are well studied. However, our contribution lies in a non-trivial integration supported by both theoretical analysis and empirical validation. Specifically, we propose a novel hyperbolic RQ-VAE framework to model a coarse-to-fine hierarchical codebook, as well as a differential-length codebook strategy to effectively compress the overall codebook size. Moreover, extensive experiments demonstrate the effectiveness of HG-Rec.
>
> ---
> Missing **Limitations** section.
>
> **Author Response:** We have carefully considered all the limitations mentioned by the four reviewers and added **Limitations and Future work**. Please refer to our response to **Missing Limitations section** raised by Reviewer **ww3f**.

---

> > ### Author Rebuttal · Reviewer_zNoz · 2026-04-04
> >
> > Thanks for the response, which has improved my understanding of this work. I will adjust the score accordingly.

---

> > > ### Author Response · Authors · 2026-04-04
> > >
> > > We sincerely thank reviewer **zNoz** for the time, effort, and positive feedback. We are pleased that our responses have addressed your concerns, and we greatly appreciate your decision to raise the rating. In the final version, we will modify the above issues, placing greater emphasis on enhancing the description and analysis of hyperbolic RQ-VAE, and presenting the configurations of codebook size in the main body instead of Appendix F.1 to improve clarity.

---

### Official Review · Reviewer_ww3f · 2026-03-20

**Soundness:** 3
**Presentation:** 3
**Significance:** 3
**Originality:** 3
**Overall Recommendation:** 4
**Confidence:** 4

**Summary:**

This paper looks at a fundamental research problem in generative recommendation - codebook-based RQ-VAE design. The authors point out the issues of current RQ-VAE due to the operation in Euclidean space. It theoretically shows that RQ-VAE induces a tree-like structure and that hyperbolic space has exponential capacity growth. Based on these properties, it proposes hyperbolic RQ-VAE with a differential-length codebook to align representation with hierarchy. Experiments show that the method improves codebook utilization, reduces collision, and enhances recommendation performance.

**Compliance With Llm Reviewing Policy:**

Affirmed.

**Key Questions For Authors:**

Q1. The paper shows that hyperbolic RQ-VAE leads to more uniform codebook utilization. However, during downstream generative recommendation training, only the discrete codes are used, rather than the continuous embeddings. How does the improved distribution of latent representations translate into better recommendation performance?
Q2. The paper claims that Euclidean space distorts hierarchical relationships. Can the authors provide more intuitive examples or theoretical justification to support this claim?

**Limitations:**

It would be good if the authors could discuss the potential limitation introduced by the manually defiend size for each layer's code sizes.

**Strengths And Weaknesses:**

**Strengths**:

S1. The paper addresses an important and fundamental problem in generative recommendation - item tokenization, which has direct impact on downstream performance.
S2. The motivation is clear and well-supported by theoretical insights, including the tree structure induced by RQ-VAE and the exponential capacity of hyperbolic space.
S3. The proposed method is reasonable and well-designed, combining hyperbolic geometry with a codebook with different sizes at each layer.
S4. The empirical evaluation is comprehensive, covering recommendation accuracy, codebook utilization, collision rate, and training efficiency, and shows consistent improvements.

**Weakness**:
W1. Although the authors theoretically show the tree sturcture property of RQ-VAE and the exponential capacity of hyperbolic space, the reason why euclidean space will distort hierarchical relationships is not theoretically justified and lacks deeper analysis beyond intuition and empirical observations.
W2. The paper does not clearly explain how improved codebook utilization in the latent space translates to better downstream recommendation performance.

---

> ### Author Rebuttal · Authors · 2026-03-30
>
> We sincerely appreciate the valuable suggestions provided by Reviewer **ww3f**. We present our point-by-point responses to the comments as follows:
>
> ---
> **W1 & Q2**: The problem of **Euclidean space distorts tree-like structure**.
>
> **Author Response:** We provide a theoretical justification for this claim.
>
> **Theorem 1: Euclidean space distorts tree-like structure**.
>
> *Proof:* Assume there exists an embedding $f: T_k \to \mathbb{R}^d $ that preserves all edge lengths to be 1 for a complete binary tree of depth $k$. Specifically, the binary tree has $2^k$ leaf nodes and the distance between any two leaf nodes is larger than 2. Make a sphere with a radius of 1 centered on each leaf node, and these $2^k$ spheres do not intersect with each other. In addition, the distance from each leaf node to the root is exactly $k$, hence all leaf nodes lie within a large ball centered at $f(\text{root})$ with radius $k + 1$. We obtain the following volume inequality: $2^k \cdot C_d \cdot 1^d \leq C_d \cdot (k+1)^d$, where $C_d$ is the volume of the unit ball in $\mathbb{R}^d$. It can be simplified as $2^k \leq (k+1)^d$. Taking logarithms: $k \leq d \cdot \log(k+1), d \geq \frac{k}{\log(k+1)}$. Since $k = \log_2 n $, this implies that the dimension $d \geq \Omega\ \left(\dfrac{\log n}{\log \log n}\right) $. Under fixed dimensions, the above inequality cannot hold. Q.E.D.
>
> ---
> **W2 & Q1**: The problem of **codebook utilization**.
>
> **Author Response:** Since discrete codes are derived from continuous embeddings in the latent space via hyperbolic residual quantization operations, they inherently contain the underlying semantic information and coarse-to-fine hierarchical structure of items. Hence, we adopt discrete codes rather than continuous embeddings for downstream GR tasks.
>
> To clearly explain how codebook utilization affects the performance of downstream GR tasks, we conduct a group of experiments on three datasets and the detailed results are presented in **Tables 1** and **2**. We construct codebooks with collision rates of around 80%, 60%, 40%, and 20% during the training of both HG-Rec and TIGER.
>
> Table 1: HG-Rec
>
> ||Beauty|Instruments|Yelp|
> |-|-|-|-|
> |Collision Rate|R@5|R@5|R@5|
> |80%|0.0417|0.0867|0.0209|
> |60%|0.0429|0.0913|0.0228|
> |40%|0.0460|0.0936|0.0243|
> |20%|0.0523|0.0956|0.0261|
> |Best|0.0572|0.1058|0.0275|
>
> Table 2: TIGER
>
> ||Beauty|Instruments|Yelp|
> |-|-|-|-|
> |Collision Rate | R@5| R@5| R@5|
> |80%|0.0400|0.0891|0.0173|
> |60%|0.0415|0.0901|0.0197|
> |40%|0.0449|0.0924|0.0213|
> |20%|0.0482|0.0947|-|
> |Best|0.0502|0.0979|0.0234|
>
> 1. Under the condition of optimal collision rate, HG-Rec is superior to TIGER. The main reason is that Hyperbolic RQ-VAE is able to learn continuous embeddings with discriminative and hierarchical characteristics.
> 2. The performance of HG-Rec consistently outperforms TIGER under different settings, representing that effective modeling hierarchical relationships across codebook layers can improve performance under codebook collapse.
> 3. As the collision rate increases, the performance of HG-Rec and TIGER gradually decreases. This is because the increasing collision rate limits the representational capability of the codebook, and a small number of codewords are frequently used. Consequently, this increases information loss and reduces discriminability, degrading downstream GR performance.
>
> ---
> Missing **Limitations** section.
>
> **Author Response:** To make our paper more complete, we have added this section and the detailed description is as follows,
>
> **Limitations**: One potential limitation of our proposed HG-Rec is that the codebook size for each layer needs to be manually specified for each dataset. Furthermore, the size of each codebook is predefined rather than adaptively learned from data, leading to suboptimal allocation of representational resources. Moreover, we incorporated hyperbolic RQ-VAE to generate a codebook with hierarchical characteristics. Although this does not introduce the extra time cost of downstream tasks, enabling downstream GR models to understand the coarse-to-fine hierarchical characteristics remains a challenge. Finally, for real-world deployment of recommendation systems, hyperbolic RQ-VAE adopts Poincaré distance to measure the similarities between codewords, which may lead to the problems of unstable training (e.g., numerical instability near the boundary of the Poincaré ball) and high training costs (e.g. expensive hyperbolic operations). Hence, it may limit the potential for large-scale deployment to some extent.
>
> **Future work**: In future work, we will explore contrastive learning techniques to better align the codebook representations with downstream task objectives. Moreover, we plan to incorporate weakly supervised learning strategies to enhance hyperbolic RQ-VAE, which can effectively alleviate the problem of original semantic forgetting. Finally, we will investigate replacing the Poincaré model with the Lorentz model to improve training stability and speed.

---

### Decision · Program_Chairs · 2026-04-30

**Decision:**

Accept (regular)

**Comment:**

All reviewers are positive about this submission, with a broad agreement on the importance of the underlying problem, the soundness of the technical design, and the empirical effectiveness. The rebuttal further addressed the concerns raised during review by providing substantial additional evidence. I therefore support a clear acceptance and strongly encourage the authors to incorporate the additional experiments and clarifications from the rebuttal and discussion into the camera-ready version.